# R1-RANKER: TEACHING LLM RANKERS TO REASON

## ABSTRACT

Large language models (LLMs) have recently shown strong reasoning abilities in domains like mathematics, coding, and scientific problem-solving, yet their potential for ranking tasks, where prime examples include retrieval, recommender systems, and LLM routing, remains underexplored. Ranking requires complex reasoning across heterogeneous candidates, but existing LLM-based rankers are often domain-specific, tied to fixed backbones, and lack iterative refinement, limiting their ability to fully exploit LLMs' reasoning potential. To address these challenges, we propose R1-Ranker, a reasoning-incentive framework built on reinforcement learning, with two complementary designs: DRanker, which generates full rankings in one shot, and IRanker, which decomposes ranking into an iterative elimination process with step-wise rewards to encourage deeper reasoning. We evaluate unified R1-Rankers on nine datasets spanning recommendation, routing, and passage ranking, showing that IRanker-3B consistently achieves state-of-the-art performance, surpasses larger 7B models on some tasks, and yields a 15.7% average relative improvement. Ablation and generalization experiments further confirm the critical role of reinforcement learning and iterative reasoning, with IRanker-3B improving zero-shot performance by over 9% on out-of-domain tasks and reasoning traces boosting other LLMs by up to 22.87%. These results demonstrate that unifying diverse ranking tasks with a single reasoning-driven foundation model is both effective and essential for advancing LLM reasoning in ranking scenarios.

## 1 INTRODUCTION

Large language models (LLMs) have recently emerged as powerful reasoners in the text space, where tasks such as mathematics (Ahn et al., 2024; Zhang et al., 2024; Ma et al., 2025), coding (Yang et al., 2024b; Zhang et al., 2025), and scientific problem-solving (Rueda et al., 2025; Wysocki et al., 2024) can be framed as structured reasoning over natural language representations. Building on this progress, it is natural to ask whether other text-centric tasks can similarly benefit from LLM reasoning. Ranking tasks, including information retrieval (Nogueira & Cho, 2019; Khattab & Zaharia, 2020), recommender systems (Cremonesi et al., 2010; He et al., 2017), and LLM routing (Li et al., 2023b; Zhang et al., 2023), are compelling, as they require reasoning across heterogeneous candidates, weighing contextual signals, and making comparative judgments (Li et al., 2023a; Qin et al., 2023). Yet, despite their significance, ranking tasks have received far less attention in the LLM reasoning literature. Therefore, our paper aims to raise attention to this pressing research question: *can we teach LLM-based text rankers to reason effectively?*

Existing LLM-based text rankers (Sun et al., 2023; Yoon et al., 2024; Hou et al., 2024b; Li et al., 2023a; Qin et al., 2023) leverage the reasoning and instruction-following capabilities of large language models to directly generate candidate rankings in a unified text-based format, *e.g.*, GPT4Rec (Li et al., 2023a) explores the use of generative LLMs for relevance ranking in information retrieval tasks, and LLM4Ranking (Liu et al., 2025) provides a framework that enables users to adopt various ranking methods using open-source or API-based LLMs for document reranking tasks. However, existing text rankers are often domain-specific (e.g., passage ranking or recommendation), largely rely on fixed LLM backbones, and lack mechanisms for iterative refinement. These limitations constrain the potential of LLMs to fully exploit their reasoning capabilities in ranking scenarios.

Constructing an ideal reasoning-incentive text ranker is a non-trivial task, which mainly involves two challenges: *(1) Building a foundation model that generalizes across domains.* An ideal ranking foundation model should be able to reason about diverse ranking tasks in a unified manner, capturing the common principles behind relevance, preference, and prioritization. Such a model would eliminate the need for task-specific customization, reduce implementation complexity, and more importantly,

Figure 1: **Example ranking tasks that a proposed R1-Ranker can solve.** (a) The recommendation task aims to model the user's preferences based on their historical behaviors. It will rank the current item candidates and predict which items the user is most likely to prefer. (b) The routing task is to recommend suitable LLMs to respond to different user queries. The recommendation process takes into account the effectiveness and cost of each LLM's response, and performs ranking to generate the final recommendation list. (c) Passage ranking involves retrieving a set of passages from candidate passages based on a given user query for retrieval-augmented generation. It ranks the passages by modeling the relevance between the query and the passages to produce the final list of passages.

unlock the potential of LLMs to generalize their reasoning across domains. *(2) Adapting LLMs in the post-training phase.* Current state-of-the-art LLMs are not explicitly optimized for ranking, even though ranking fundamentally requires decision-making capabilities. While API-based adaptations have shown promise, their effectiveness remains constrained by the inherent limitations of the underlying LLMs. We argue that ranking should be incorporated into the post-training "recipe" of LLMs to unlock their full potential for reasoning in ranking scenarios.

In this paper, we introduce R1-Ranker, a reasoning-based LLM ranker built on a reinforcement learning (RL) framework, which is designed to reason about diverse ranking tasks in a unified manner. R1-Ranker includes two variants, DRanker and IRanker, which offer different perspectives on incorporating reasoning into ranking. We first consider a basic design, DRanker, which directly leverages the LLM's reasoning ability to generate a complete candidate ranking and then optimizes it with RL based on task-specific rewards. While intuitive and able to exploit the LLM's basic reasoning ability for ranking decisions, this approach requires the model to directly rank multiple candidates, leading to an excessively large output space and limited room for deeper reasoning within the LLM's context window. To address these limitations, we further propose IRanker, a refined framework that integrates RL with iterative decoding. Instead of directly generating the entire ranking, IRanker decomposes the task into a step-wise elimination process: the model incrementally reasons about candidate quality and excludes the worst candidate from the pool until the final order is obtained by reversing the exclusion sequence. This design dramatically reduces the output space, alleviates context-length constraints, and enables more deliberate reasoning during training.

We meticulously train and evaluate unified R1-Rankers on nine representative datasets spanning three scenarios: recommendation, routing, and passage ranking. Our results show that a single IRanker-3B not only matches or surpasses domain-specific methods for each task but also achieves state-of-the-art (SOTA) performance compared to general ranker baselines, highlighting the effectiveness of reasoning-based approaches for ranking. Remarkably, IRanker-3B even outperforms larger 7B LLMs on some tasks and achieves a 15.7% relative improvement on average. We further conduct extensive ablation studies, which confirm that both our RL design and the iterative mechanism are key to stimulating deeper reasoning and ensuring robustness across different LLM sizes. In addition, both in-domain and out-of-domain zero-shot generalization experiments demonstrate the transferability of reasoning for ranking: IRanker-3B improves over the base model by at least 5% on in-domain ranking tasks, and by over 9% on out-of-domain tasks such as GSM8K, IFEval, and MathQA. Notably, we show that the intermediate reasoning traces generated during training further enhance the zero-shot ranking capabilities of other LLMs. For example, based on the 3B backbone model, these reasoning traces can lead to a 23% relative improvement in zero-shot performance for the base model. In sum, our key contributions are: (1) a single R1-Ranker that works well across recommendation, routing, and retrieval without retraining or task-specific designs; (2) a novel iterative decoding algorithm that simplifies LLM output space for efficient reasoning with limited context window.

## 2 PRELIMINARIES

**Ranking tasks**. Given a query $q$, a ranking task (Liu et al., 2009; Li, 2011; Cao et al., 2007) is to learn a ranker $f$ that ranks a set of candidates $D = \{c_1, \ldots, c_n\}$ with $n$ elements. The candidate set is typically divided into a positive candidate set $D_p$ and a negative candidate set $D_n$. The positive

candidate set refers to the items selected by the user, for example, the items a user purchased in a recommendation system. To evaluate how well the ranker recovers these positive candidates, the performance of the ranking task is measured by an evaluator $E$, which is usually a ranking metric such as Mean Reciprocal Rank (MRR) (Voorhees et al., 1999). The goal of the ranker $\pi$ is to learn a function that maps a query $q$ and its candidate set $D$ to a ranking order $O = \{c_1^{r_1}, c_2^{r_2}, \ldots, c_n^{r_n}\} \in \mathbb{S}_n$ with $r_i$ as the rank of candidate $c_i$, such that the evaluation metric $E$ is maximized:

$$\pi : (q, D) \to O, \quad O \in \mathbb{S}_n, \tag{1}$$

where $\mathbb{S}_n$ denotes the set of all possible permutations over $n = |D|$ elements. Formally, the optimal ranker $\pi^*$ is learned by solving:

$$\pi^* = \arg \max_{f \in \mathcal{F}} \mathbb{E}_{(q,D)\sim\mathcal{Z}} \left[ E(\pi(q, D)) \right], \tag{2}$$

where $\mathcal{F}$ is the function class and $\mathcal{Z}$ is the data distribution over queries and candidate sets.

**Examples**. As shown in Figure 1, we have listed some representative instantiations that can be unified into a ranking foundation model from three aspects. **(a) Recommendation**, shown in Figure 1(a), the goal of the recommendation (Ricci et al., 2010; Adomavicius & Tuzhilin, 2005; Covington et al., 2016) is to capture user preferences by analyzing their histories, rank the current set of candidate items, and predict those that the user is most likely to favor. Here, the user history is modeled as query $q$, each element of the candidates is an item, and positive candidate set $D_p$ contains the real items that the user would choose. **(b) Routing**, routing task (Ong et al., 2024; Feng et al., 2024; Huang et al., 2025; Hu et al., 2024) aims to recommend appropriate LLMs for handling diverse user queries by ranking them based on both effectiveness and response cost, shown in Figure 1(b). The final LLM recommendation list is generated through a ranking process that balances performance with efficiency. Specifically, a user query is regarded as query $q$, each element of the candidates is a LLM name or LLM description (Feng et al., 2024), and the positive candidate set $D_p$ means the ground truth LLMs for the query. **(c) Passage ranking**, passage ranking (Guu et al., 2020; Karpukhin et al., 2020; Lewis et al., 2020) aims to identify and reorder the most relevant passages from a set of candidates given a user query, often arises in retrieval-augmented generation, shown in Figure 1(c). This is achieved by modeling the relevance between the query and each passage to produce a final ranking list. For this task, the user query is regarded as query $q$, each element of the candidates is a passage, and the positive candidate set $D_p$ contains the ground truth passages.

## 3 DRANKER: TRAINING A BASIC TEXT-RANKER TO REASON WITH RL

### 3.1 TRAINING AND INFERENCE VIA RL

We leverage RL to enhance the reasoning ability of DRanker $\pi_\theta$, enabling it to not only optimize candidate rankings based on performance feedback from the evaluator $E$ but also to progressively refine its reasoning strategies across diverse ranking tasks.

**Ranking reward**. We first model the evaluator $E$ as a ranking reward $r_a = E(O_d)$, with the objective of maximizing $E$. Specifically, since the number of candidates varies across different ranking tasks, we choose MRR as the evaluator $E$. However, due to the limited instruction-following ability of LLMs, the output candidate rankings often miss or add candidates, which violates the standard requirements of a valid ranking. To regularize the output of the DRanker and guide more faithful reasoning, we introduce a format reward as a penalty term $r_g = \Omega(O_d, D) - 1$, where $\Omega$ calculates the F1 score measuring the overlap between $O_d$ and $D$. Thus, the final reward is defined as $r_d = r_a + r_g$, combining both task-specific ranking quality and reasoning faithfulness to the input candidate set.

**Training and inference via PPO**. To optimize the reasoning-driven DRanker $\pi_\theta$ for maximizing the final reward $r_d$, we adopt the Proximal Policy Optimization (PPO) algorithm (Schulman et al., 2017), a widely used actor-critic RL method for LLMs. PPO trains the policy by maximizing the objective:

$$\mathcal{J}_{\text{PPO}}(\theta) = \mathbb{E}_{(q,D)\sim\mathcal{Q}, \, y\sim\pi_\theta^{\text{old}}(\cdot|P_t(q,D))} \left[ \sum_{t=1}^{|y|} \min \left( \frac{\pi_\theta(y_t \mid P_t(q,D), y_{<t})}{\pi_\theta^{\text{old}}(y_t \mid P_t(q,D), y_{<t})} A_t, \right. \right.$$
$$\left. \left. \text{clip} \left( \frac{\pi_\theta(y_t \mid P_t(q,D), y_{<t})}{\pi_\theta^{\text{old}}(y_t \mid P_t(q,D), y_{<t})}, 1 - \epsilon, 1 + \epsilon \right) A_t \right) \right] \tag{3}$$

where $\pi_\theta$ and $\pi_\theta^{\text{old}}$ denote the current and previous policies, respectively. The pair $(q, D) \sim \mathcal{Q}$ is sampled from the training distribution, where $q$ denotes the input query and $D$ the associated

Figure 2: **Framework of our proposed R1-Ranker.** Both DRanker and IRanker are RL-enhanced LLM frameworks that exploit the reasoning ability of LLMs to solve ranking tasks. They take as input the candidate information in text form, along with user information (such as user history or a query), and utilize LLM reasoning to produce a final candidate ranking. This ranking is then evaluated by an evaluator to generate a corresponding reward signal, which is used to optimize the decision-making of both rankers. The key distinctions are: 1) DRanker performs reasoning once to generate the full ranking in a single step, whereas IRanker conducts step-wise reasoning by iteratively excluding the least likely item from the candidate pool. 2) The reward in DRanker is a ranking reward based on the final candidate list, while the reward in IRanker is an exclusion reward provided for each individual decision, which encourages finer-grained reasoning. 3) DRanker always receives the full set of candidates as input with a fixed size, whereas IRanker's input candidates are dynamically updated based on the excluded items, enabling adaptive reasoning throughout the ranking process.

candidate set. $y = \{y_1, y_2, \ldots, y_{|y|}\}$ is the token-level sequence generated by the policy model. This sequence is parsed into a ranking candidate list $O_d = \texttt{Parse}(y, D)$ using a matching function based on string similarity or identifier alignment with $D$. The advantage estimate $A_t$ is computed using Generalized Advantage Estimation (GAE) (Schulman et al., 2015), based on future rewards $\{r_{\geq t}\}$ and a learned value function $V_\phi$. The clipping parameter $\epsilon$ is used to ensure stable policy updates. In this way, PPO not only stabilizes learning but also encourages the policy to explore more effective reasoning trajectories for ranking. After obtaining the optimal $\pi_\theta^*$, we can obtain the final ranking candidate list $O_d = \texttt{Parse}(\pi_\theta^*(P_t(q, D)), D)$.

# 4 IRANKER: ADVANCED TEXT RANKER WITH RL AND ITERATIVE DECODING

## 4.1 DECOMPOSE CANDIDATES RANKING INTO SINGLE CANDIDATE SELECTIONS

Although DRanker can rank candidates in a simple and direct manner, it is limited by the large action space and the constrained context length for reasoning. To address this problem, we propose IRanker, as shown in the lower part of Figure 2. The key insight of IRanker is to decompose the complex global ranking task into a sequence of localized reasoning steps, where the model iteratively excludes one candidate from the pool. This formulation transforms ranking into an incremental decision-making process that not only reduces the combinatorial complexity but also allows the LLM to perform more fine-grained reasoning at each step. Similar to the method introduced in Section **??**, we design a prompt $P_d$ (details in Appendix A) that encodes the query $q$ and the current candidate set $D^{(k)}$, and instructs the IRanker $\pi_\omega$ to reason about candidate quality and exclude one candidate $c_k \in D^{(k)}$ at each iteration $k$. The process is defined as:

$$c_k = \pi_\omega(P_d(q, D^{(k)})), \quad D^{(k-1)} = D^{(k)} \setminus \{c_k\}, \quad r_k = |D| - k + 1 \quad (4)$$

After $|D|$ iterations, we obtain the complete ranking list:

$$O_d = [c_1^{r_1}, c_2^{r_2}, \ldots, c_{|D|}^{r_{|D|}}] \quad (5)$$

Table 1: **Detailed summarization of tasks used in our ranking tasks.** We summarize the task names, scenarios, candidate sizes, training/test case counts, and positive/negative ratios.

| Dataset | Scenario | Candidate Size | # Train Cases | # Test Cases | Positive/Negative Ratio |
|---------|----------|----------------|---------------|--------------|-------------------------|
| Movie | Recommendation | 20 | 9975 | 2508 | 1:19 |
| Music | Recommendation | 20 | 9975 | 2508 | 1:19 |
| Game | Recommendation | 20 | 9424 | 2337 | 1:19 |
| Performance | Routing | 10 | 1467 | 153 | 1:9 |
| Cost | Routing | 10 | 1467 | 153 | 1:9 |
| Balance | Routing | 10 | 1467 | 153 | 1:9 |
| 5 Passage | Passage ranking | 5 | 10000 | 1448 | 1:4 |
| 7 Passage | Passage ranking | 7 | 9996 | 3894 | 1:6 |
| 9 Passage | Passage ranking | 9 | 10000 | 1792 | 1:8 |

Here, candidates excluded later receive higher ranks, and the final ranking $O_d$ is constructed by reversing the exclusion order. This iterative exclusion mechanism provides a natural way for the LLM to reason step by step, narrowing down the candidate space progressively instead of making all decisions at once.

### 4.2 TRAINING AND INFERENCE WITH ITERATIVE RL

**Exclusion reward**. To train this iterative reasoning process, we first define the exclusion reward $r_e^k$ for each step $k$:

$$r_e^k = \begin{cases} 1, & \text{if } c_k \in D_n \\ 0, & \text{otherwise} \end{cases} \tag{6}$$

where $D_n$ is the negative candidate set illustrated in Section 2. The exclusion reward encourages IRanker $\pi_\omega$ to reason about relevance at each step by prioritizing the removal of negative candidates, thereby promoting positive candidates to be ranked higher.

**Training and inference via iterative PPO**. Similar to Section 3.1, we utilize PPO to optimize the reasoning policy of IRanker $\pi_\omega$ by maximizing the following objective:

$$\begin{aligned} \mathcal{J}_{\text{PPO}}(\omega) = \mathbb{E}_{(q, D^{(k)}) \sim \mathcal{B}, \, y \sim \pi_\omega^{\text{old}}(\cdot \mid P_d(q, D^{(k)}))} &\left[ \sum_{t=1}^{|y|} \min \left( \frac{\pi_\omega(y_t \mid P_d(q, D^{(k)}), y_{<t})}{\pi_\omega^{\text{old}}(y_t \mid P_d(q, D^{(k)}), y_{<t})} A_t, \right.\right. \\ &\left.\left. \text{clip}\left( \frac{\pi_\omega(y_t \mid P_d(q, D^{(k)}), y_{<t})}{\pi_\omega^{\text{old}}(y_t \mid P_d(q, D^{(k)}), y_{<t})}, 1 - \epsilon, \, 1 + \epsilon \right) A_t \right) \right] \end{aligned} \tag{7}$$

where $(q, D^{(k)}) \sim \mathcal{B}$ is sampled from the training distribution; $P_d(q, D^{(k)})$ is the prompt encoding the query $q$ and current candidate pool $D^{(k)}$; $\pi_\omega$ and $\pi_\omega^{\text{old}}$ denote the current and previous policies, respectively; $A_t$ is the estimated advantage at decoding step $t$; $\epsilon$ is the PPO clipping threshold; $y = (y_1, y_2, \ldots, y_T)$ is the response sequence generated by the policy; $c_k = \texttt{Parse}(y, D^{(k)})$ is the excluded candidate parsed from $y$. By optimizing this objective, PPO encourages IRanker to develop consistent reasoning strategies across iterations. Once the optimal $\pi_\omega^*$ is obtained, the final ranking list $O_d$ is constructed via Equation (5) by repeatedly applying Equation (4) with $\pi_\omega^*$.

## 5 EXPERIMENTS

We conduct comprehensive training and evaluation of the proposed R1-Ranker, DRanker and IRanker. Notably, **the same R1-Ranker is being evaluated across diverse 9 interdisciplinary tasks**, which is compared against general ranking methods and domain-specific methods. First, we introduce the tasks within the R1-Ranker framework.

**Task description**. The details of the tasks are summarized across three aspects in Table 1. **(1) Recommendation (Rec)**: For the recommendation ranking task, we utilize three widely-used sequential recommendation datasets: MovieLens ml-1m (Hou et al., 2024a), Amazon's CD and Vinyl dataset (McAuley et al., 2015; Ni et al., 2019), and Amazon's Video Game dataset (McAuley et al., 2015; Ni et al., 2019). For each user across all datasets, following the settings of (Hou et al., 2024a), we extracted 20 consecutive interactions as the historical sequence and designated the 21st interaction as the ground truth item. To create a balanced candidate set, we randomly sampled 19 items from the complete item catalog (excluding items in the user's history and the ground truth item)

Table 2: **Model performance comparison with general baselines across nine ranking tasks of three scenarios on MRR**. **Bold** and underline denote the best and second-best results. We can observe the following: 1) Compared to the baselines, *IRanker-3B* achieves state-of-the-art performance in almost all tasks. 2) The comparison between methods with and without RL validates the enhancement effect of RL on ranking tasks. 3) The comparison between iterative-based ranking and direct ranking demonstrates the suitability of the iterative design for models of different sizes.

| Model | Recommendation | | | Routing | | | Passage Ranking | | |
|---|---|---|---|---|---|---|---|---|---|
| | Movie | Music | Game | Performance | Cost | Balance | 5 Passages | 7 Passages | 9 Passages |
| *Retrieval-based Models* | | | | | | | | | |
| *BM25* | 17.56 | 18.09 | 14.96 | 18.41 | 13.52 | 13.39 | 53.63 | 44.95 | 39.69 |
| *Contriever* | 18.29 | 17.04 | 23.98 | 20.75 | 16.29 | 16.74 | 41.91 | 36.41 | 33.10 |
| *Representative Text Rankers* | | | | | | | | | |
| *GPT4Rec* | 23.62 | 24.53 | 15.86 | 10.39 | 13.96 | 11.30 | 28.89 | 24.37 | 21.81 |
| *PRP* | 27.30 | 16.52 | 30.52 | 19.52 | 16.97 | 18.50 | 45.67 | 30.67 | 17.72 |
| *Direct-Rank LLMs without RL* | | | | | | | | | |
| *Qwen2.5-3B-Instruct-direct* | 16.92 | 16.68 | 13.17 | 10.00 | 10.00 | 10.00 | 38.08 | 22.47 | 15.94 |
| *Qwen2.5-7B-Instruct-direct* | 16.59 | 17.29 | 18.63 | 13.38 | 13.51 | 18.41 | 44.57 | 23.69 | 17.79 |
| *Iterative LLMs without RL* | | | | | | | | | |
| *Qwen2.5-3B-Instruct-iter* | 22.01 | 21.97 | 29.49 | 20.87 | 20.22 | 12.42 | 57.74 | 43.47 | 39.40 |
| *Qwen2.5-7B-Instruct-iter* | 22.11 | 23.36 | 33.14 | 19.13 | 21.06 | **26.09** | **62.01** | 50.94 | 48.74 |
| *Direct-Rank LLMs with RL* | | | | | | | | | |
| *DRanker-3B* | 18.71 | 15.70 | 15.77 | 20.63 | 9.06 | 13.38 | 43.85 | 22.86 | 16.11 |
| *Iterative LLMs with RL* | | | | | | | | | |
| *IRanker-3B* | **34.69** | **29.18** | **42.49** | **23.62** | **30.39** | 24.44 | 60.98 | **53.22** | **49.96** |

and combined them with the ground truth item to form a candidate list of 20 items. This approach creates a realistic recommendation scenario with a 5% chance of randomly selecting the relevant item, while maintaining a manageable evaluation space. **(2) Routing (Router)**: For the LLM routing task, following the setting of (Feng et al., 2024), we selected four datasets from (Feng et al., 2024) and ten large language models (LLMs). Based on different weights of LLM response effectiveness and cost, we followed three settings in (Feng et al., 2024): Performance First (Performance), Balance, and Cost First (Cost), corresponding to scenarios where users prioritize high performance, value both high performance and low cost equally, or prioritize low cost, respectively. For each query, we computed the reward based on the weighted sum of effectiveness and cost, and selected the LLM with the highest reward as the ground-truth LLM. The remaining nine LLMs were treated as negative LLMs. **(3) Passage Ranking (Passage)**: For the passage ranking task, we employ the MS MARCO passage dataset (Bajaj et al., 2016), a large-scale information retrieval benchmark derived from Bing search logs. To evaluate models under different retrieval complexity scenarios, we created three distinct settings with varying candidate passage sizes: 5, 7, and 9 passages per query. For each query, these candidate sets include one relevant passage (as judged by human annotators in the dataset) and 4, 6, or 8 irrelevant passages respectively. This configuration allows us to assess how model performance scales with increasing candidate pool sizes and how effectively models can identify the single relevant passage among varying numbers of distractors.

**Baselines and metrics**. We evaluate a variety of baseline methods across three scenarios. The baselines are categorized into two groups: **(a) *General baselines*** that apply across tasks, and **(b) *Task-specific baselines*** tailored to each scenario. For all methods, we primarily use Mean Reciprocal Rank (MRR) (Voorhees et al., 1999; Cremonesi et al., 2010) to evaluate ranking performance in the main text. A full evaluation with additional metrics and generation cases are provided in Appendix D and B for details. **(a) *General baselines***: We consider three categories: retrieval-based, text rankers, and LLM-based methods. For retrieval, we adopt *BM25* (Robertson et al., 2009), a classical probabilistic retrieval model with keyword matching, and *Contriever* (Izacard et al., 2021), a dense retriever trained with contrastive learning and hard negatives. For text rankers, we include *GPT4Rec* (Li et al., 2023a), which reformulates recommendation as a text-to-text generation task using LLMs with task-specific prompts, and *PRP* (Qin et al., 2023), which casts ranking as pairwise preference prompting and aggregates LLM-comparisons to form the final list. For LLM-based baselines, we use *Qwen2.5-3B/7B-Instruct* (Yang et al., 2024a), and design two variants: *LLM-direct*, which produces rankings in a single step, and *LLM-iter*, which performs iterative candidate selection for improved accuracy. **(b) *Task-specific baselines***: For recommendation, we compare three sequential models: *SASRec* (Kang & McAuley, 2018), a Transformer-based sequential recommender; *BPR* (Rendle et al.,

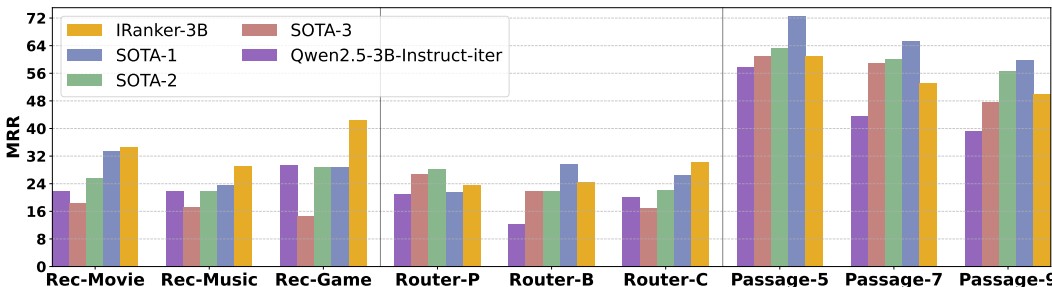

Figure 3: **IRanker-3B matches the performance of domain-specific methods across multiple tasks with fewer training samples and smaller model size.** We compared the performance of IRanker-3B against three representative SOTA methods and Qwen2.5-3B-Instruct-iter across three scenarios. SOTA-1, SOTA-2, and SOTA-3 correspond to SASRec (Kang & McAuley, 2018), BPR (Rendle et al., 2012), and R1-Rec (Lin et al., 2025) in the recommendation (Rec) scenario; GraphRouter (Feng et al., 2024), RouterBert (Ong et al., 2024), and RouterKNN (Hu et al., 2024) in the routing (Router) scenario; RankLLama-8B (Ma et al., 2024), RankBERT (Nogueira & Cho, 2019), and MonoT5 (Nogueira et al., 2020) in the passage ranking (Passage) scenario.

Table 3: **Zero-shot performance comparison across different ranking tasks on MRR**. **Bold** and underline denote the best and second-best results. The results for each ranking scenario were obtained by training on the data from the other two ranking scenarios and then performing zero-shot testing on the target scenario.

| | Recommendation | | | Routing | | | Passage Ranking | | |
|---|---|---|---|---|---|---|---|---|---|
| Model | Movie | Music | Game | Performance | Cost | Balance | 5 Passages | 7 Passages | 9 Passages |
| Qwen2.5-3B-Instruct-iter | 22.01 | 21.97 | 29.49 | 20.87 | 20.22 | 12.42 | 57.74 | 43.47 | 39.40 |
| IRanker-3B | **34.69** | **29.18** | **42.49** | **23.62** | **30.39** | **24.44** | **60.98** | **53.22** | **49.96** |
| IRanker-3B (zero-shot) | 25.95 | 23.21 | 31.16 | 20.41 | 23.10 | 21.89 | 56.42 | 51.19 | 42.45 |

2012), which optimizes pairwise item preferences; and *R1-Rec* (Lin et al., 2025), a reinforcement learning framework that optimizes retrieval-augmented LLMs via task feedback. For routing, we use *RouterKNN* (Hu et al., 2024), which assigns queries by nearest-neighbor voting; *RouterBERT* (Ong et al., 2024), a lightweight BERT-based classifier for routing decisions; and *GraphRouter* (Feng et al., 2024), a state-of-the-art graph-based router balancing accuracy and cost. For passage ranking, we evaluate *RankBERT* (Nogueira & Cho, 2019), a BERT reranker fine-tuned on MS MARCO; *MonoT5* (Nogueira et al., 2020), which generates relevance labels with a T5 model; and *RankLLama-8B* (Ma et al., 2024), a Llama-2 variant fine-tuned with pairwise and listwise objectives for passage ranking.

**Implementation details**. We train and evaluate a single R1-Ranker across all tasks, comparing its performance against both general-purpose ranking baselines and domain-specific methods tailored for each task. For reinforcement learning, we adopt Proximal Policy Optimization (PPO) (Schulman et al., 2017), following the implementation details provided in VeRL (Sheng et al., 2024). Both the DRanker and IRanker are initialized from Qwen-2.5-3B-Instruct, and optimized using KL-regularized policy gradients. To manage policy divergence, we incorporate a low-variance KL loss with a regularization coefficient of 1e-4. Rollouts are conducted using vLLM with a maximum GPU memory utilization cap of 40%, temperature set to 0.9, and a maximum response length of 1024 tokens. Training is performed over 5 epochs, using a learning rate of 1e-6 for the actor and 2e-6 for the critic. The optimization uses a global mini-batch size of 36 and a micro-batch size of 8. To ensure memory efficiency, we enable gradient checkpointing and apply Fully Sharded Data Parallelism (FSDP) with both parameter and gradient offloading. All experiments are conducted on NVIDIA A6000 GPUs.

## 5.1 IRANKER ACHIEVES SOTA PERFORMANCE COMPARED WITH GENERAL RANKING METHODS

We train a unified IRanker-3B on all tasks and test it across all tasks without further fine-tuning, compared with general ranking methods. We report the comparison results in Table 4.2. We can observe that: (1) IRanker-3B surpasses all baseline models in the majority of ranking tasks. Notably, it even outperforms the larger Qwen2.5-7B-Instruct-iter model in 7 out of 9 tasks, despite having fewer parameters. On average, IRanker-3B achieves a 15.7% relative improvement in MRR over

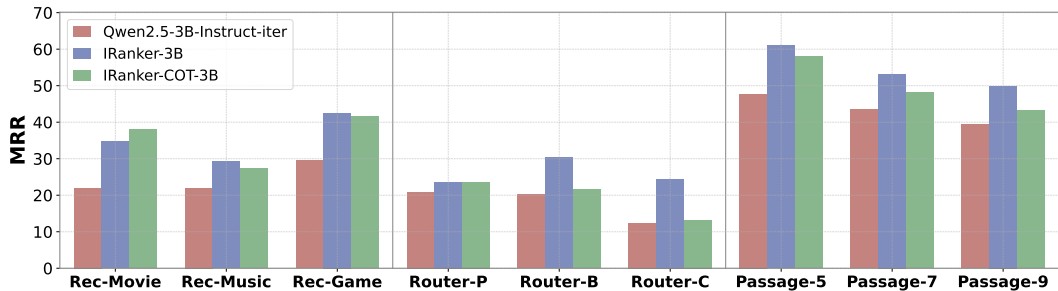

Figure 4: **Thoughts emerged by IRanker during training can enhance zero-shot performance of the base model.** IRanker-COT-3B is an iterative framework that, for each test query, retrieves similar queries and their corresponding thoughts that emerged during the training of IRanker, using them as thought templates to guide zero-shot responses. We evaluate IRanker-COT-3B on nine tasks and compare its performance with IRanker-3B and Qwen2.5-3B-Instruct-iter. The results show that IRanker-COT-3B consistently outperforms Qwen2.5-3B-Instruct-iter and even surpasses IRanker-3B in the Rec-Game task.

Qwen2.5-7B-Instruct-iter across the nine evaluated tasks, showing that reasoning-oriented ranking design can achieve superior efficiency and accuracy compared to scaling alone. (2) The performance gap between models with and without reinforcement learning (RL), particularly when comparing IRanker-3B to its counterparts without RL, confirms that RL plays a critical role in stimulating deeper reasoning for ranking. For example, IRanker-3B significantly improves over Qwen2.5-3B-Instruct-iter in routing tasks (e.g., Cost: 30.39 vs. 20.22) and recommendation tasks (e.g., Game: 42.49 vs. 29.49), demonstrating that RL not only improves optimization but also enhances the model's ability to reason about candidate relevance. (3) When comparing direct ranking models to their iterative counterparts, iterative designs (e.g., Qwen2.5-7B-Instruct-iter and IRanker-3B) consistently outperform direct methods across all task domains. This suggests that decomposing global ranking into step-wise elimination enables more deliberate reasoning at each decision step, leading to stronger task alignment and more robust generalization across tasks of varying complexity.

## 5.2 IRANKER MATCHES THE PERFORMANCE OF DOMAIN-SPECIFIC METHODS ACROSS MULTIPLE TASKS

We further compared IRanker-3B with domain-specific methods as shown in Figure 3. Specifically, we compared IRanker-3B with three representative SOTA methods (denoted as SOTA-1, SOTA-2, and SOTA-3) and Qwen2.5-3B-Instruct-iter across three domains. In the recommendation scenario, SOTA-1, SOTA-2, and SOTA-3 refer to SASRec, BPR, and R1-Rec, respectively; in routing, they correspond to GraphRouter, RouterBert, and RouterKNN; and in passage ranking, they denote RankLLama-8B, RankBERT, and MonoT5. We can observe that: (1) IRanker-3B matches the performance of domain-specific methods across multiple tasks, showing that unified reasoning-driven ranking can rival highly specialized models. (2) In the recommendation scenario, IRanker-3B outperforms all baselines and achieves state-of-the-art results, suggesting that iterative reasoning allows the model to capture user–item relevance signals more effectively than handcrafted domain-specific designs. (3) Even in the passage ranking scenario, IRanker-3B performs on par with models that have significantly larger parameter sizes (e.g., RankLLama-8B) or were trained with much more data samples (e.g., RankBERT and MonoT5). This highlights that reasoning-centric optimization can serve as a competitive alternative to brute-force scaling or domain-specific engineering.

## 5.3 IRANKER ENHANCES ZERO-SHOT REASONING FOR RANKING TASKS AND TRANSFERS TO BROADER DOMAINS

To further investigate the generalization ability of IRanker as a reasoning-driven foundation model, we conduct analyses from the following three aspects.

**IRanker strengthens reasoning transfer across in-domain ranking tasks**. We first examine the ability of IRanker to generalize reasoning strategies across ranking tasks, by training it on two different ranking tasks and then conducting zero-shot testing on a target ranking task (Table 3). IRanker-3B (zero-shot) consistently outperforms Qwen2.5-3B-Instruct-iter and even approaches the performance of fully trained IRanker on tasks such as Performance and 7 Passages. This shows that

Table 4: **IRanker outperformed the base model on three out-of-domain generic LLM tasks.** Bolded values indicate higher performance. This table compares the performance of IRanker-3B and Qwen2.5-3B-Instruct across eight widely-used benchmarks. IRanker-3B leads in five out of eight tasks, especially on math and reasoning-heavy datasets like GSM8K, IFEval, and MathQA. Qwen2.5-3B-Instruct performs better on code generation tasks, including MBPP and HumanEval. The models are nearly tied on general QA tasks like OpenBookQA and HellaSwag. These results highlight IRanker-3B's strength in structured reasoning, while Qwen2.5-3B-Instruct maintains a slight edge in coding ability.

| Model | GSM8K | IFEval | MMLU | MBPP | HumanEval | OpenBookQA | HellaSwag | MathQA |
| Metric | Exact Match Acc. | Loose Acc. | Acc. | Pass@1 | Pass@1 | Acc. | Acc. | Acc. |
|---|---|---|---|---|---|---|---|---|
| Qwen2.5-3B-Instruct | 0.6353 | 0.6799 | **0.6537** | **0.5280** | **0.4756** | 0.3280 | 0.5633 | 0.3538 |
| IRanker-3B | **0.7369** | **0.7122** | 0.6510 | 0.4560 | 0.4573 | **0.3300** | **0.5634** | **0.3856** |

the reasoning patterns learned during training can be effectively transferred across ranking tasks, leading to strong in-domain generalization.

**Emergent reasoning traces boost other LLMs' zero-shot capability**. To investigate whether the intermediate reasoning traces generated by IRanker can further enhance the base model, we propose IRanker-COT-3B. This model adopts an iterative framework that, for each test query, retrieves semantically similar queries along with their reasoning traces produced during IRanker training. These retrieved thoughts serve as templates (Appendix C) to guide zero-shot inference. As shown in Figure 4, IRanker-COT-3B consistently outperforms Qwen2.5-3B-Instruct-iter and even exceeds IRanker-3B on the Rec-Game task, highlighting the transferable value of reasoning traces.

**IRanker improves reasoning ability on out-of-domain tasks**. Finally, we test whether IRanker's reasoning capability transfers beyond ranking by evaluating IRanker-3B and Qwen2.5-3B-Instruct on eight generic LLM benchmarks (Table 4). IRanker-3B outperforms Qwen2.5-3B-Instruct on five of the eight tasks, with notable gains on reasoning-intensive datasets such as GSM8K, IFEval, and MathQA. Conversely, Qwen2.5-3B-Instruct remains stronger on code generation tasks (MBPP and HumanEval), while performance is comparable on general QA tasks (OpenBookQA and HellaSwag). These results demonstrate that IRanker is particularly effective at structured reasoning, extending its benefits beyond ranking scenarios.

## 6 ADDITIONAL RELATED WORK

Generative large language models (LLMs) have recently been applied to ranking tasks across diverse domains such as information retrieval, recommendation, and document reranking, leveraging their natural language understanding and generation capabilities. Prompting-based methods (Qin et al., 2023; Hou et al., 2024a) exploit LLM generalization to produce rankings with minimal modification, while instruction tuning and alignment approaches such as GPT4Rec (Li et al., 2023a) and RankRAG (Yu et al., 2024) further adapt LLMs to ranking-specific signals. Beyond these, reinforcement learning has emerged as a powerful tool to enhance reasoning in ranking, with methods like Rank-R1 (Zhuang et al., 2025) and Rec-R1 (Lin et al., 2025) demonstrating that reward-driven optimization can align LLM outputs with downstream ranking objectives and improve performance in both general and personalized scenarios.

## 7 CONCLUSION

In this work, we address the challenge of unifying diverse ranking tasks by introducing IRanker, an iterative R1-Ranker optimized via reinforcement learning. By decomposing ranking into a step-wise exclusion process and leveraging the reasoning capabilities of large language models, IRanker overcomes limitations of traditional embedding-based and direct-ranking methods. Our proposed IRanker-3B achieves competitive or state-of-the-art performance across nine datasets from recommendation, routing, and passage ranking scenarios. Extensive experiments demonstrate its strong generalization abilities, achieving over 5% improvement in in-domain zero-shot settings and over 9% gains in out-of-domain LLM tasks. These results highlight IRanker's effectiveness as a unified and scalable R1-Ranker framework, setting a foundation for future advances in LLM-based ranking systems with wide applications in recommendation, retrieval, and decision making.

ETHICS STATEMENT

All authors of this paper have read and adhered to the ICLR Code of Ethics. Our work does not involve human subjects, personal data, or sensitive attributes. We followed best practices for data usage, ensured compliance with licensing terms, and considered potential risks of bias or misuse.

REPRODUCIBILITY STATEMENT

We have made every effort to ensure the reproducibility of our results. Details of the model architecture, training settings, and hyperparameters are described in Section 5. All datasets we used are publicly available. The training scripts and evaluation code will be released upon publication to facilitate replication.

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

## A  PROMPT USAGE

This section provides a detailed overview of the prompt templates used for each task scenario, corresponding to DRanker and IRanker. Specifically, $P_t$ denotes the prompt template used for DRanker, and $P_d$ refers to the template used for IRanker. Each prompt is carefully designed with explicit formatting instructions and consistently requires the model to articulate its reasoning process before producing a final answer, which is enclosed within `<answer>` and `</answer>` tags for ease of parsing. Illustrative examples of the $P_t$ templates are shown in Tables 5, 6, and 7, while representative $P_d$ templates are presented in Tables 8, 9, and 10.

Table 5: **Prompts for DRanker in recommendation task**.

```
<|im_start|>
```
*system*
You are a helpful assistant that ranks products by how likely the user is to buy them, based on their previous purchase history.
```
<|im_end|>
<|im_start|>
```
*user*
I've purchased the following items in the past, in order:
**{historical_interactions}**
Now there are 20 candidate items that I might purchase next:
**{candidate_items}**
Please rank these items by measuring the possibilities that I would like to buy next most, according to my purchase history. Please think step by step.
Split your output with line break. You MUST rank the given candidate items. You can not generate items that are not in the given candidate list. Show your work in `<think>` `<think>` tags. And return the final answer in `<answer>` `</answer>` tags.
```
<|im_end|>
<|im_start|>
```
*assistant*
Let me solve this step by step.
```
<think>
```

## B  CASE STUDIES OF DRANKER AND IRANKER

This appendix presents a comprehensive set of case studies illustrating the behavior of **DRanker** and **IRanker** across different ranking scenarios. For each model, we provide detailed examples from three distinct tasks, each further divided into three subtasks, resulting in **nine case tables per ranker**.

Each case study table includes the following components:

- **Question:** The input query or prompt that initiates the ranking task.
- **Positive Sample:** The ground-truth relevant passage or item that should ideally be highly ranked.
- **Answer:** The model's output, including its ranked list and any intermediate reasoning (when applicable).

These examples serve to highlight both the strengths and failure modes of each ranker, offering qualitative insights into their decision-making processes. DRanker case studies are provided in Tables 11 to 19, while IRanker case studies are shown in Tables 20 to 28.

## C  EXAMPLES OF THOUGHT TEMPLATE GENERATED BY IRANKER

This section presents examples of *thought templates* generated by IRanker and DRanker's thinking process, each summarizing the model's step-by-step reasoning for representative queries. These templates are derived from model inference traces and can be used as in-context learning exemplars for models not explicitly trained on the ranking tasks. See Tables 30, 29 and 31

Table 6: **Prompts for DRanker in routing task**.

```
<|im_start|>
```
*system*
You are a helpful assistant that selects the most suitable large language model (LLM) for a given query, based on performance and token cost.
```
<|im_end|>
```
```
<|im_start|>
```
*user*
**{llm_descriptions}**
## This scenario is analyzing a set of sales data to uncover trends and insights. Please provide useful insights with reasonable depth, balancing accuracy and efficiency. Here is a query: **{query}** and LLM candidates: **{llm_candidates}**. Please think step by step according to the description of each query and LLM, and evaluate from the perspectives of performance in answering the query and token price. Rank all LLMs from most suitable to least suitable for this query. Return the LLM names in order, one per line. Split your output with line break. You MUST rank all LLMs from the candidate list. You can not generate content that is not in the given candidate list.
Show your work in `<think> <think>` tags. And return the final answer in `<answer> </answer>` tags.
```
<|im_end|>
```
```
<|im_start|>
```
*assistant*
Let me solve this step by step.
```
<think>
```

Table 7: **Prompts for DRanker in passage ranking task**.

```
<|im_start|>
```
system
You are a helpful assistant that ranks passages by relevance to a given query.
```
<|im_end|>
```
```
<|im_start|>
```
user
## Here is a query: **{query}**
**{formatted_passages}**
Please think step by step according to the content of each passage and how well it supports or relates to the query. Rank all passages from most relevant to least relevant. Return the passage IDs in order, one per line (e.g.,
passage 1
passage 3
passage 2). You MUST rank all passages from the candidate list. You can not generate content that is not in the given candidate list.
Show your work in `<think> <think>` tags. And return the final answer in `<answer> </answer>` tags.
```
<|im_end|>
```
```
<|im_start|>
```
assistant
Let me solve this step by step.
```
<think>
```

# D    ADDITIONAL RESULTS ON EXTENDED METRICS

This section presents supplementary results using additional evaluation metrics including nDCG@k. The detailed outcomes can be found in Tables 32, 33, and 34. While zero-shot performance can be found in Tables 35, 36, and 37.

Table 8: **Prompts for IRanker in recommendation task**.

```
<|im_start|>
```
*system*
You are a helpful assistant that ranks products by how likely the user is to buy them, based on their previous purchase history.
```
<|im_end|>
<|im_start|>
```
*user*
I've purchased the following items in the past, in order:
**{historical_interactions}**
Now there are 20 candidate items that I might purchase next:
**{candidate_items}**
Please select the one item that is least likely to be my next purchase, according to my purchase history. Please think step by step. You MUST choose exactly one item from the given candidate list. You can NOT generate or reference items that are not in the given candidate list. Show your work in `<think>` `<think>` tags. And return the final answer in `<answer>` `</answer>` tags.
```
<|im_end|>
<|im_start|>
```
*assistant*
Let me solve this step by step.
```
<think>
```

Table 9: **Prompts for IRanker in routing task**.

```
<|im_start|>
```
*system*
You are a helpful assistant that selects the most suitable large language model (LLM) for a given query, based on performance and token cost.
```
<|im_end|>
<|im_start|>
```
*user*
**{llm_descriptions}**
## This scenario is analyzing a set of sales data to uncover trends and insights. Please provide useful insights with reasonable depth, balancing accuracy and efficiency. Here is a query: **{query}** and LLM candidates: **{llm_candidates}**. Please think step by step according to the description of each query and LLM, and evaluate from the perspectives of performance in answering the query and token price, and select the least likely LLM from the LLM candidates. Only return the LLM name corresponding to the LLM. You MUST choose one LLM name from LLM candidates. You can not generate content that are not in the given LLM candidates. Show your work in `<think>` `<think>` tags. And return the final answer in `<answer>` `</answer>` tags.
```
<|im_end|>
<|im_start|>
```
*assistant*
Let me solve this step by step.
```
<think>
```

# E  BROADER IMPACTS

Our work on IRanker, a unified ranking foundation model, has several potential positive societal implications. By creating a single model capable of handling multiple ranking tasks across recommendation systems, LLM routing, and passage retrieval, we significantly increase efficiency and reduce computational resources needed across various applications. This could lead to more sustainable AI

Table 10: **Prompts for IRanker in passage ranking task**.

```
<|im_start|>
```
*system*
You are a helpful assistant that ranks passages by relevance to a given query.
```
<|im_end|>
<|im_start|>
```
*user*
## Here is a query: **{query}**
**{formatted_passages}**
Please think step by step according to the content of each passage and how well it supports or relates to the query. Select the least likely passage from the candidate list. Only return the passage ID corresponding to the excluded passage (e.g., "passage 3"). You MUST choose one passage from the candidate list. You can not generate content that is not in the given candidate list.
Show your work in `<think> <think>` tags. And return the final answer in `<answer>`
`</answer>` tags.
```
<|im_end|>
<|im_start|>
```
*assistant*
Let me solve this step by step.
```
<think>
```

deployment and democratize access to high-quality ranking technologies for smaller organizations with limited resources. The improved zero-shot capabilities demonstrated by IRanker could enhance information retrieval in low-resource domains or for underrepresented languages where task-specific training data is scarce. This has the potential to bridge information access gaps across different communities. Furthermore, IRanker's ability to perform well on out-of-domain tasks suggests that the techniques developed in this work may have beneficial spillover effects to other AI domains beyond ranking. Our iterative decoding approach, which decomposes complex ranking tasks into simpler decisions, represents a more interpretable way of understanding how AI systems make ranking decisions. This improved transparency could foster greater trust in recommendation and information retrieval systems. Additionally, the efficient use of context length in our approach enables more effective reasoning with limited computational resources, potentially reducing the environmental footprint of deploying such systems at scale while maintaining high performance.

## F  THE USE OF LARGE LANGUAGE MODELS (LLMS)

An LLM was used during the preparation of this manuscript to improve the clarity of writing. Specifically, the tool assisted in correcting grammatical errors, refining sentence structures, and polishing the overall readability of some passages. All substantive ideas, analyses, and conclusions are solely those of the authors.

Table 11: **DRanker's case study in rec-movie scenario.**

**Question**: ['0. My Fair Lady', '1. Top Hat', '2. Next Karate Kid, The', '3. Gone with the Wind', '4. Dave', '5. Raiders of the Lost Ark', '6. Englishman Who Went Up a Hill, But Came Down a Mountain, The', '7. Star Trek: First Contact', '8. Star Trek: The Motion Picture', '9. Star Trek VI: The Undiscovered Country', '10. Star Trek V: The Final Frontier', '11. Star Trek: The Wrath of Khan', '12. Star Trek III: The Search for Spock', '13. Star Trek IV: The Voyage Home', '14. Star Trek: Insurrection', '15. Star Trek: Generations', '16. From Russia with Love', '17. Goldfinger', '18. Star Wars: Episode I - The Phantom Menace', '19. Star Wars: Episode IV - A New Hope']

**Ground Truth**: Lassie

`<think>` First, I will analyze each genre to determine which types of movies align with those previously viewed:

**Historical / Period Films**:
 '0. My Fair Lady', '3. Gone with the Wind', '9. Star Trek: The Undiscovered Country'
**Hollywood Musicals**:
 '1. Top Hat', '11. William Shakespeare's Romeo and Juliet'
**Action**:
 '6. The Englishman Who Went Up a Hill, But Came Down a Mountain',
 '19. Star Wars: Episode IV - A New Hope', '18. Star Wars: Episode I - The Phantom Menace',
 '1.1 Star Trek: First Contact', '13. Star Trek: The Wrath of Khan'
**Classics**:
 '2. The Next Karate Kid', '16. From Russia with Love', '17. Goldfinger'
**Modern Classics**:
 '4. Dave', '2. West Side Story', '5. Raiders of the Lost Ark'

Based on this analysis, I grouped the movies into genre categories. Now I'll predict the most likely category for your next movie choice:
- Given the chronological pattern in your movie consumption, it's likely that upcoming selections will continue from the genres you've recently explored.
- You also appear to be a fan of Westerns or horseback-themed films. Movies such as *'Tashunga'*, which blends Western themes with action, support this trend.
- Your interest in *'Shakes the Clown'*—a unique Western-style Elvis movie—suggests a preference for genre blends that combine entertainment and classic elements.
- Among your top movies, *'Frankenstein Meets the Wolfman'* (a part of the Universal Monsters collection) reveals your appreciation for B-movies with gothic or Poe-like undertones.
This pattern suggests your next movie choice will likely fall into a Western, Classic, or genre-blended category, with potential interest in entertainment-leaning or horror-inspired classics.
`</think>`
`<answer>` 1. 'Tashunga'
2. 'Clean Slate'
3. 'West Side Story'
4. 'Lassie'
5. 'From Russia with Love'
6. 'Home for the Holidays'
7. '8MM'
8. 'Cemetery Man'
9. '16. From Russia with Love'
10. 'Shakes the Clown'
11. 'Untouchables, The'
12. 'West Side Story'
13. 'Beauty and the Beast'
14. 'Beauty and the Beast'
15. 'The Friggerito'
16. 'Lassie'
17. 'Clean Slate'
18. 'West Side Story'
`</answer>`

Table 12: **DRanker's case study in rec-music scenario.**

**Question**: ['0. Horny House', '1. Mozart: Piano Sonatas KV 282, 310 & 545', '2. Mozart: Complete Piano Trios', '3. Mozart: Piano Sonatas KV. 281/KV. 330/KV. 333/Rondo, KV. 485/Adagio, KV. 540', '4. Mozart: The Sonatas for Piano and Violin', '5. Beethoven: Piano Sonatas No. 8, Pathétique/ No. 14, Moonlight/ No. 21, Waldstein/ No. 23, Appassionata', '6. Mozart: Violin Concertos Complete', '7. You Took My Heart By Surprise', '8. Revenge', '9. Prokofiev: Romeo & Juliet - Scenes from the Ballet', '10. Ian Bostridge - The English Songbook', '11. Hawaiian Slack Key Christmas', '12. The Sinatra Christmas Album - Chorus and Orchestra Conducted By Gordon Jenkins', '13. Niki Nana', '14. Distant Fields', '15. Whenever You Need Somebody', '16. Free', '17. Somewhere My Love', '18. Shared Vision: Songs of the Beatles', '19. American Live Debut']

**Ground Truth**: Classical Hits

`<think>` First, it's important to analyze the relationship between the albums I have purchased in the past and the genres represented in the current list. To do this, I'll group the albums by musical genre and determine how likely it is that my future selections will align with those genres.

**Genre Clusters:**

*Classical*:
  'Horny House', 'Mozart: Piano Sonatas', 'Mozart: Complete Piano Trios', 'Mozart: Piano Sonatas', 'Mozart: Piano Sonatas', 'Mozart: Violin Concertos Complete', 'Beethoven: Piano Sonatas No. 8, Pathétique / No. 14, Moonlight / No. 21, Waldstein / No. 23, Appassionata'

*Pop/Rock*:
  'Niki Nana', 'Somewhere My Love', 'American Live Debut', 'Shared Vision: Songs of the Beatles'

*Alternative / New Music*:
  'World of Absence', 'Fabriclive 54 :'

*Jazz*:
  'Big Band = Jazz'

The most frequently purchased genres are Classical and Pop/Rock, suggesting a strong preference. Therefore, future purchases are likely to align with these categories.

Non-classical albums such as *'Wild World'* are more thematic and require additional context to fully assess their appeal. Similarly, *'World of Absence'* is described as an ambient album, which, while intriguing, stands apart from the dominant genres.

Albums like *'Timeagain'*, *'Now'*, and *'Pop: Remixed'* show a focus on distinct thematic or remix elements, which may still capture my interest, albeit less frequently.

In summary, judging from past purchases, Classical and Pop genres are the most likely candidates for future music selections. This aligns well with my historical preferences.
`</think>`
`<answer>` Classical Hits
2. Timeagain
3. Motif
4. Diamond Life
5. Now
6. Original Good Ol' Boy
6. Niki Nana
7. Shared Vision: Songs of the Beatles
7. Now He Sings Now He Sobs
8. Lone Justice
9. A Broken Frame
10. World of Absence
11. Fabriclive 54 :
12. Pop: Remixed
13. Upon This Rock
14. Wild World
15. I Don't Want to Be
`</answer>`

Table 13: **DRanker's case study in rec-game scenario.**

**Question**: ['0. WWE 2K15 - PS3 [Digital Code]', '1. WWE 2K15 - PS3 [Digital Code]', '2. NBA 2K15 - PlayStation 3', '3. Madden NFL 15: 2,200 Points - Xbox One Digital Code', "4. Injustice: Gods Among Us - Collector's Edition - Playstation 3", "5. Injustice: Gods Among Us - Collector's Edition - Playstation 3", "6. Injustice: Gods Among Us - Collector's Edition - Playstation 3', '7. LEGO: Marvel - PC', '8. Madden NFL 15: 2,200 Points - Xbox One Digital Code', '9. Lego: Marvel Super Heroes, XBOX 360', '10. WB Games Middle Earth: Shadow of Mordor - Playstation 4', '11. The Amazing Spider-Man 2 - Nintendo 3DS', '12. Teenage Mutant Ninja Turtles - Nintendo Wii', '13. Disney Epic Mickey 2: The Power of Two - Playstation 3', '14. Disney Epic Mickey 2: The Power of Two - Playstation 3', "15. Assassin's Creed IV Black Flag - PC", '16. Dead Rising 3', '17. NBA 2K14 - Xbox 360', '18. Grand Theft Auto V: Premium Edition - Xbox One [Digital Code]', '19. NBA 2K14 - Xbox 360']

**Ground Truth**: Call of Duty Ghosts - PS3 [Digital Code]

`<think>` Firstly, I need to understand patterns in my historical purchases. I have bought several games over time, including franchise titles such as *WWE*, *NBA*, *Madden NFL*, and *Injustice: Gods Among Us*. I also seem to favor unique items, like *WWE* digital codes, which appeared in different variants multiple times.

On the PC side, I am drawn to games that offer more freedom, such as *LEGO: Marvel* and *The Amazing Spider-Man 2*. This suggests a preference for open-world or exploratory gameplay on that platform.

Next, I'll consider how the sequence of past purchases might influence future decisions. Since I've consistently purchased games, it is likely that my next purchase will also be a game. This reflects a strong continuity in gaming content consumption.

Additionally, I've noticed some imbalance between console and PC purchases. To address this, I may lean toward buying more individual titles rather than digital codes or bundles to diversify my collection.

Finally, considering my history of PC-centric games and smaller discrete items like accessories, these categories could also shape my upcoming choices.
`</think>`
`<answer>` 1. Call of Duty Ghosts - PS3 [Digital Code]
2. Rise of the Tomb Raider - Xbox One Digital Code
3. Xbox One S Vertical Stand
4. Hermitshell Hard Travel Case for 8Bitdo Sn30 Pro Bluetooth Gamepad (SN30PRO-GB)
5. Mind's Eye: Secrets of the Forgotten [Download]
6. MLB 2K16 - Xbox One Digital Code
7. Microsoft Combat Flight Simulator 2: Pacific Theater - PC
8. The Amazing Spider-Man 2 - Nintendo 3DS
9. Family Feud [Download]
10. Amazing Match 3 Games - Volume 3
11. Phineas and Ferb: Across the 2nd Dimension - Playstation 3
12. Evil Dead: A Fistful of Boomstick
13. Turtle Beach - Ear Force PX51 Wireless Gaming Headset - Dolby Digital - PS3, Xbox 360
14. Red Black & Blue Silicone Skin Case Combo for PS3 Controller
15. Gabriel Knight Mysteries: Limited Edition - PC
16. Family Feud [Download]
17. Untold Legends: Brotherhood of the Blade - Sony PSP
18. Batman: The Animated Series
19. NBA 2K16 - Xbox One Digital Code
20. WB Games Middle Earth: Shadow of Mordor - Playstation 4
`</answer>`

Table 14: **DRanker's case study in router-performance scenario.**

| |
|---|
| **Question**: Explain why the given definition is wrong.A mole is an animal that lives underground. |
| **Ground Truth**: Llama-3.1-Turbo (70b) |

`<think>` The task here is to critique a definition and provide insights with reasonable depth. The given definition is "A mole is an animal that lives underground. " From the description provided, LLaMA-3 (8b) and Mistral-7b are the two models that mentions less capacity for detailed explanations and comprehensive insights due to their lower token price and model sizes.

Mixtral-8x7B was described as excelling at instruction-based tasks, step-by-step reasoning, solid reading comprehension, and structured summaries. NousResearch (34b) was ideal for complex research queries, multi-step reasoning, deep reading comprehension, and thorough summaries. LLAma-2 (7b) has similar token price and capacity to LLaMA-3 (8b) and is lightweight for straightforward reasoning and comprehension. Qwen-1. 5 (72b) is described as versatile for challenging reasonsing, extended reading comprehension, and comprehensive summaries.

Given the need to find a model that can provide detailed, clear insights and reasoning, Qwen-1. 5 (72b) stands out as the most suitable for the task given its higher token price and better description. Next, NousResearch (34b) would be suitable for producing detailed reasonings with its higher capacity as well. LLaMA-3 (8b) and Mistral-7b are less suitable as they have limitations in capacity also with lower token prices. Lastly, Mixtral-8x7B could also be a suitable model, but LLaMA-3-Turbo (70b) and LLaMA-3-Turbo (8b) are better choices given their upgrades in capacity and token prices. LLaMA-3-Turbo (70b) specifically deals with strong reasoning validation, which fits well with the essence of the query. Therefore, we're left with only one model for comprehensive reasoning and analyses - Qwen-1. 5 (72b). `</think>`
`<answer>` Qwen-1. 5 (72b)
LLaMA-3-Turbo (70b)
NousResearch (34b)
LLaMA-3-Turbo (8b)
`</answer>`

Table 15: **DRanker's case study in router-balance scenario.**

| |
|---|
| **Question**: How many teams have played in the Super Bowl eight times? |
| **Ground Truth**: LLaMA-3-Turbo (8b) |

`<think>` For this query, we're looking for information where the answer is likely to be a specific number (a count of teams) rather than a detailed explanation of a process or interpretation of data. The information required is factual but not too advanced, as there is no need for extensive reasoning or deep reading comprehension here. We need a straightforward and concise response using LLMs.

Given the description, the most suitable LLMs would be those in the categories of *Lightweight* and *Fast*, as they are best at simple Hybrid QA and short reading comprehension tasks, ensuring a brief and efficient answer.

Starting with the LLMs:

- **LLaMA-3 (8b)** - Lightweight chat model for straightforward Hybrid QA with concise summaries which is well-suited for this straightforward query.

- **Mistral-7b** - Fast for moderate Hybrid QA with brief summaries, consistent with the straightforward nature of the query.

- **LLaMA-2 (7b)** - Similar in lightweight category, but more moderate in reasoning and responses versus the more detailed LLaMA-3. It is still suitable for our task.

- **Qwen-1.5 (72b)** - Versatile for challenging Hybrid QA. Although it's described as versatile, this level of versatility isn't essential for a simple query like this and it could be a bit overpowered.

- The other LLMs, like **Mixtral-8x7B**, **NousResearch (34b)**, **LLaMA-3 (70b)**, **LLaMA-3-Turbo (70b)**, and **LLaMA-3.1-Turbo (70b)**, all indicate more advanced reasoning, which is not necessary for this query.

`</think>`
`<answer>` LLaMA-3 (8b)
Mistral-7b
LLaMA-2 (7b)
`</answer>`

Table 16: **DRanker's case study in router-cost scenario.**

| |
|---|
| **Question**: How many fumbles did Von Miller force in Super Bowl 50? |
| **Ground Truth**: LLaMA-3-Turbo (8b) |

`<think>` First, we need to determine the suitability of each LLM for the given query. The query is quite straightforward, asking for a specific statistic (number of fumbles) specifically about Von Miller in a Super Bowl.

Now, we'll look at the requirements and capabilities mentioned for each of the available LLMs in relation to simple fact queries:

1. **LLaMA-3 (8b):** Simplified information, extremely short comprehensions, and brief summaries.

2. **Mixtral-8x7B:** Instruction-based tasks, step-by-step reasoning, solid reading comprehension, and structured summaries.

3. **NousResearch (34b):** Comprehensively searches through complex questions, but assumes advanced understanding.

4. **LLaMA-2 (7b):** Moderately deeper understanding, yet fairly limited reasoning among others.

5. **Mistral-7b:** Moderate Hybrid QA, short reasoning time, and brief summaries.

6. **LLaMA-3 (70b):** Advanced Hybrid QA, deep reasoning, detailed comprehension, and extensive summaries.

7. **LLaMA-3-Turbo (8b):** Balanced and reasonable reasoning time, moderate summarization.

8. **LLaMA-3-Turbo (70b):** Powerful, strong reasoning, deep reading comprehension, and high-quality summaries.

9. **LLaMA-3.1-Turbo (70b):** Specific to instructional tasks, thorough reasoning, structured reasoning.

10. **Qwen-1.5 (72b):** For challenging tasks, versatile reasoning, extended reading comprehension, and comprehensive summaries.

Given the simple nature of the query, **LLaMA-3 (8b)** is most suitable as the information retrieval and fact-checking might be straightforward for it. However, since the query is fact-dependent and balances between a high-quality summary and accuracy across simple fact-checks, LLaMA-3 (8b) might struggle with accuracy due to its limitations in handling complex reasoning.
**Mixtral-8x7B** and **LLaMA-3.1-Turbo (70b)**, on the other hand, seem well-suited as they can handle concise and brief information retrieval, though they are likely not strong in advanced reading or fact extraction.
**LLaMA-3-Turbo (8b)** and **LLaMA-3-Turbo (70b)** may be marginal choices given their balance of performance across reasoning, summaries, and fact retrieval.
**Mistral-7b** is not the best choice as it is mostly focused on efficiency rather than balance.
**LLaMA-3 (70b)** is a powerful model but better suited for more complex tasks and reasoning.
**NousResearch (34b)** and **Qwen-1.5 (72b)** could also handle the query but may not provide the level of detail and accuracy required for this specific fact-checking task.
Hence, the rough order of suitability for this query might be:

- **Mistral-7b** (efficient)

- **LLaMA-3-Turbo (8b)** (balanced)

- **LLaMA-3-Turbo (70b)** (advanced)

`</think>`
`<answer>` Mistral-7b
LLaMA-3-Turbo (8b)
LLaMA-3-Turbo (70b)
`</answer>`

Table 17: **DRanker's case study in passage-5 scenario.**

| |
|---|
| **Question**: How much to print at staples? |
| **Ground Truth**: passage 1 |
| `<think>` First, I will analyze each passage to determine how well it supports or relates to the query "how much to print at Staples". |
| Passage 1: This passage discusses the cost variability of printing at Staples but does not provide specific pricing details. |
| Passage 2: This passage provides specific pricing for black and white copies, which is relevant to the query. |
| Passage 3: This passage highlights benefits like clean, crisp copies and low pricing recommendations, but does not directly address pricing at Staples. |
| Passage 4: This passage includes specific pricing for both black and white and color copies at Staples, making it highly relevant. |
| Now, I will rank the passages according to their relevance: |
| - Passage 4 provides the most detailed pricing information. |
| - Passage 2 gives specific pricing for black and white copies. |
| - Passage 1 discusses cost variability without giving exact prices. |
| - Passage 3 is the least relevant, as it does not provide any specific pricing details. |
| `</think>` |
| `<answer>` passage 4 |
| passage 2 |
| passage 1 |
| passage 3 |
| `</answer>` |

Table 18: **DRanker's case study in passage-7 scenario.**

| |
|---|
| **Question**: How long leave cooked chicken at room temperature? |
| **Ground Truth**: passage 6 |
| `<think>` First, I will analyze each passage to determine their relevance to the query about cooked chicken and room temperature. The query asks how long cooked chicken can be kept at room temperature and what action should be taken. |
| All passages discuss food safety related to perishable items like chicken. While several mention the danger of room temperature exposure, I will prioritize those that clearly state a specific time limit and recommended actions. |
| Passages 0 through 3 reiterate general safety concerns but lack a specific timeframe. |
| Passage 4 mentions the two-hour limit (also noted in Passage 3) but does not clearly recommend refrigeration or freezing. |
| Passage 5 is the most directly relevant, clearly stating that cooked chicken should not remain at room temperature for more than two hours. |
| Passage 6 also provides full guidance, including the two-hour limit and the recommendation to freeze the chicken, making it highly informative. |
| `</think>` |
| `<answer>` passage 5 |
| passage 6 |
| passage 4 |
| passage 3 |
| passage 0 |
| passage 1 |
| passage 2 |
| `</answer>` |

Table 19: **DRanker's case study in passage-9 scenario.**

| |
|---|
| **Question**: How much weight can a floor joist support? |
| **Ground Truth**: passage 5 |
| `<think>` Firstly, we need to establish which information from the given passages is directly relevant to the weight capacity of a floor joist. |
| Passage 5 is highly relevant, as it calculates the weight a 3-foot by 12-foot section of flooring can support based on standard floor loads. |
| Passage 2 references weight in pounds but focuses on joist dimensions, which is less directly relevant than explicit weight calculations. |
| Passage 7 provides the most direct answer by specifying the weight a single floor joist can support in load-bearing situations. |
| Passage 3 supports the calculations presented in Passage 5, confirming its accuracy. |
| Passage 6 essentially duplicates the relevant content from Passage 5 using nearly identical language. |
| Passage 8 discusses general safety factors, which are important contextually but do not directly answer the query. |
| Passage 1 contains technical material that does not pertain specifically to joist weight calculations. |
| Passage 4 mentions joists in the broader context of floor framing requirements, making it less relevant to our focused question. |
| `</think>` |
| `<answer>` passage 7 |
| passage 5 |
| passage 6 |
| passage 2 |
| passage 1 |
| passage 3 |
| passage 8 |
| passage 7 |
| `</answer>` |

Table 20: **IRanker's case study in rec-movie scenario.**

**Question**: ['0. Cop Land', '1. Great White Hype, The', '2. Toys', '3. Mummy, The', '4. Frighteners, The', '5. Blair Witch Project, The', '6. Pleasantville', '7. Midsummer Night's Dream, A', '8. Haunting, The', '9. Home Alone 2: Lost in New York', '10. Addams Family Values', '11. Fatal Attraction', '12. Brady Bunch Movie, The', '13. White Men Can't Jump', '14. I Still Know What You Did Last Summer', '15. Mighty Ducks, The', '16. Jingle All the Way', '17. Jack', '18. Dunston Checks In', '19. Santa Clause, The']

**Ground Truth**: Stuart Saves His Family

<think> First, let's consider the genre and style of the movies I've watched. My previous 17 movies cover a range of genres and styles:
- Action: *Cop Land, Toys, Frighteners, White Men Can't Jump, Dick Tracy, Edge, Shallow Grave*
- Comedy: *Toys, Frighteners, Pleasantville, I Still Know What You Did Last Summer, Jingle All the Way, S Curve, Dunston Checks In*
- Drama: *The Mummy, Midsummer Night's Dream, Sense and Sensibility, From the Journals of Jean Seberg, Stuart Saves His Family, Bonfire of the Vanities*
- Horror: *The Blair Witch Project, Frighteners, Halloween, Addams Family & Addams Families, Addams Family Values*
- Fantasy: *Alice in Wonderland, Santo vs. Frankenstein, Santo vs. Werewolf*
- Family: *Home Alone, The Mummy, The Christmas Rush, Under the Cherry Tree, The Santa Clause, The Vampire Bros.*
- Romance: *Pleasantville, Fatal Attraction, Shallow Grave, Six Ways to Sunday*
- Psychological Thriller: *Fatal Attraction, Frighteners, The Mummy*
- Crime: *Budgerigar, Doll Face, Star, Searchers*
- Sports: *The Mighty Ducks*
Now, let's examine my candidate list for each genre and style:
1. Action: No movies.
2. Comedy: No movies.
3. Drama: No movies.
4. Horror: *Sense and Sensibility* is the only drama.
5. Fantasy: *Desiree* is the only fantasy.
6. Family: *Unzipped, Welcome to Woop-Woop, Fred Claus* are the only family movies.
7. Romance: *Desperate* is the only romance.
8. Psychological Thriller: There are no movies that fit 'psychological thriller' perfectly, but *Jane Eyre* is a psychological drama.
9. Crime: *Unzipped*.
10. Sports: Without adding new sports movies.
Looking at this, *Unzipped* (Drama/Horror) is the most common genre among my previous movies. In contrast, among the list we're given, only *Dick Tracy* and *Stuart Saves His Family* fit the Drama and Action genres, respectively. *From the Journals of Jean Seberg* and *Sense and Sensibility* fall more in the drama range, but not in my usual combined movie tastes.
Considering the genres, *Dick Tracy* stands out as it's a good comic book buddy cop movie, but it doesn't feature the action elements in my watching history. However, since it's a new addition and not part of my past genre preferences, it's still slightly out of place. Between the two, *Stuart Saves His Family* might be slightly more out of place as it's a bit dated in terms of its style and is closer to my usual genre than *Dick Tracy*.
So, *Stuart Saves His Family* is the one that does not seem to align with my usual pattern in my watching history. </think>
<answer> Stuart Saves His Family </answer>

Table 21: **IRanker's case study in rec-music scenario.**

**Question**: ['0. Wildewoman', '1. True North', '2. Platinum & Gold Collection', '3. The Very Best of Ben E. King', '4. All My Life', '5. Opera Goes to the Movies', '6. Charo and Guitar', '7. Greatest Hits', '8. Carole King Tapestry', '9. Cheers to the Fall', "10. Ravel: Ma Mere L'oye / Bolero / la Valse / Pavane", '11. Rimski Korsakov, Borodin: Sheherazade Op.35', '12. Without You: The Best of Harry Nilsson', '13. Blue Film', '14. The Very Best Of Aaron Neville', '15. American Bach: Cello Suite No. 1 in G Major, BWV 1007; Cello Suite No. 3 in C Major, BWV 1009 / Miles: Suite for the Americas', '16. Martha Argerich: The Collection 1 - The Solo Recordings', '17. Eagles: The Very Best Of', "18. Didn't He Ramble", '19. Greatest Hits']

**Ground Truth**: Pli selon pli

`<think>` First, let's consider the patterns in your purchase history. We see that you have mostly purchased music albums by various artists like Ben E. King, Carole King, Aaron Neville, and the Eagles. Additionally, you own a few classical music albums by composers like Ravel and Rimsky-Korsakov.

Now, looking at the list of candidate items you might purchase next:

- **0. Pli selon pli** — This is a track from an electronic duo, so we don't have a past history of purchasing this type of music.
- **1. In Through the Out Door 2015** — This is also an artist-based product, but the name of the album is quite specific and not aligned with your past purchase history.
- **2. Love Deluxe** — It's another artist-based product, likely by Coldplay, and similar to your past albums.
- **3. Exposed Valerie Simpson** — This is an artist-based product that matches the pattern of your past purchases with traditional artists.
- **4. State of Emergency** — This is another artist-based product, possibly by a punk rock band, though it somewhat resembles the classical/classical fusion albums you've bought.

We need to determine which item feels the least likely based on patterns in your past purchases. Observing your history, we see a strong trend toward:

1. Popular artists' albums
2. Various classical music albums
3. Professional or well-known albums

**Pli selon pli** stands out the most as it's an electronic piece, distinctly different from what you've traditionally bought. While **Exposed Valerie Simpson** follows the same pattern as *The Very Best of Ben E. King*, the electronic nature of **Pli selon pli** makes it the least consistent with your preferences.

**Conclusion:** Since **Pli selon pli** is completely different (electronic music) from everything else in your purchase history, it appears to be the *least likely* selection, despite being a music album. `</think>`

`<answer>` Pli selon pli `</answer>`

Table 22: **IRanker's case study in rec-game scenario.**

**Question**: ['0. HDE ® Dual Charging Station Dock for 2 Sony Playstation PS3 Controllers', '1. Far Cry 4 - PS3 [Digital Code]', '2. Turtle Beach Call of Duty Advanced Warfare Ear Force Sentinel Task Force Gaming Headset for PlayStation 4', '3. Nyko Modular Charge Station (New Version) Dual Port Controller Charging Station with Patented Charging Dongles for All PS4 , PS4 Slim and PS4 Pro', '4. Wipeout 2 - Xbox 360', '5. Kinect Star Wars - Xbox 360', '6. Far Cry Primal - PlayStation 4 Standard Edition', "7. Uncharted 4: A Thief's End - PlayStation 4", '8. Battlezone - PlayStation VR', '9. Eve: Valkyrie - PlayStation VR', '10. Robinson: The Journey - PlayStation VR', '11. Dead Rising 4 - Xbox One', '12. LEGO Jurassic World - Xbox One Standard Edition', '13. Halo 5: Guardians 9 Gold REQ Packs – Xbox One [Digital Code]', '14. Sniper Elite 4 - PlayStation 4', '15. Resident Evil 7: Biohazard - PlayStation 4', '16. Corsair VOID Wireless RGB Gaming Headset, White', '17. DOOM VFR - PlayStation 4', '18. Star Wars Battlefront II - Xbox One', '19. Far Cry 5 Gold Edition - Xbox One [Digital Code]']

**Ground Truth**: Dark Tales Edgar Allan Poe THE MASQUE OF RED DEATH Hidden Object PC Game +

`<think>` First, I'll compare the purchased items to the candidate list. The purchased items are all console games, wireless headsets, and a flying motorcycle simulator. None of the candidate items include a flying motorcycle simulator, so that's out.

Next, I'll check the types of console games:

1. PS3 - I own dozens of PS3 games

2. PS4/PS4 Slim/PS4 Pro - I bought several PS4 games

3. Xbox 360/DX - I own Xbox 360 games

4. PS VR - I own PSVR games

5. Other - I don't have games for these platforms, and want to keep watching/playing the Xbox 360 games until they're no longer on sale / Xbox 360 is apt to vanish

The PSVR and Xbox 360 game categories are more diverse, with more than one of these categories in my initial purchase history. Except for the PSVR, it seems that I was more interested in PS4 and Xbox consoles. Now, among the candidate list:

1. PC Hidden Object game - This is a genre I haven't tried before. It's far more different from console games I've purchased.

2. Nintendo Switch Game - A different hardware platform. I prefer game diversity over another system.

Both candidate games are different enough types that I haven't had the chance to play them yet.

Considering diversity and my interest, PC Hidden Object appears to be the less likely purchase since it's kind of a different category from the other games. I would rather pick a new game to add to my PlayStation or Xbox, rather than start on a new gaming experience. Nintendo Switch would likely be fun, but I've seen that people who owned it tend to go back to their preferred gaming platform. Therefore, I would avoid the Nintendo Switch game.

Thus, *Dark Tales: Edgar Allan Poe – The Masque of Red Death Hidden Object PC Game + Bonus* appears unlikely to be my next purchase. `</think>`

`<answer>` Dark Tales Edgar Allan Poe THE MASQUE OF RED DEATH Hidden Object PC Game + `</answer>`

Table 23: **IRanker's case study in router-balance scenario.**

| |
|---|
| **Question**: Explain why the following fraction is equivalent to 1/4 and 4/16 |
| **Ground Truth**: NousResearch (34b) |
| `<think>` 1. The task is to analyze a set of sales data to uncover trends and insights. This likely involves complex mathematical operations and in-depth logical reasoning. The description of LLaMA-3 (8b) and LLaMA-2 (7b) mentions they are good at brief, impartial summaries, and reasoning but not highly advanced methods. The descriptions of Qwen-1.5 and Mixtral-8x7B suggest they are specialized for nuanced reasoning and multi-step tasks. NousResearch (34B) specializes in deep reading comprehension. |

2. The query provided is "Explain why the following fraction is equivalent to 1/44/16". This implies that the LLM needs to demonstrate the equivalence using mathematical methods. It is a straightforward arithmetic problem that shouldn't be overly complex.

3. We need to consider token price in our decision. The LLMs with the highest token price are Qwen-1.5 (72b) and LLaMA-3-Turbo (70b), and the LLMs with the lowest token price are LLaMA-2 (7b) and Mixtral-8x7B.

4. Looking at LLaMA-2 (7b) and Mixtral-8x7B:
   - They are described as being more suited to step-by-step reasoning and detailed reading comprehension, which might be useful.
   - The price for Mixtral-8x7B is high, but LLaMA-2 (7b) is quite affordable, so this offers a balance.

5. Looking at Qwen-1.5 (72b):
   - Despite its name, it's described as versatile for challenging Hybrid QA, which seems too specialized for a straightforward fraction equivalence problem.
   - Its token price is also very high, making it less cost-effective.

6. Considering NousResearch (34b):
   - While it's described as ideal for complex research queries and deep reading comprehension, this problem is not complex research, it's a simple fraction simplification.
   - Its token price is expensive, and it doesn't seem ideal for solving this straightforward problem.

Based on this reasoning, the least likely LLM for answering this straightforward mathematical query involving fraction equivalence, considering both performance and cost, seems to be NousResearch (34b). `</think>`

`<answer>` NousResearch (34b) `</answer>`

Table 24: **IRanker's case study in router-cost scenario.**

**Query**: # The LLM names and their descriptions are:
# LLaMA-3 (8b): Each token price is 0.2. Handles simple Hybrid QA, basic reasoning, short reading comprehension, and concise summaries.
# Mixtral-8x7B: Each token price is 0.6. Excels at instruction-based tasks, step-by-step reasoning, solid reading comprehension, and structured summaries.
# NousResearch (34b): Each token price is 0.9. Ideal for complex research queries, multi-step reasoning, deep reading comprehension, and thorough summaries.
# LLaMA-2 (7b): Each token price is 0.2. Lightweight chat model for straightforward Hybrid QA, moderate reasoning, short reading comprehension, and concise summaries.
# Mistral-7b: Each token price is 0.2. Fast for moderate Hybrid QA, quick reasoning, short reading comprehension, and brief summaries.
# LLaMA-3 (70b): Each token price is 0.9. High-capacity model for advanced Hybrid QA, deep reasoning, detailed reading comprehension, and extensive summaries.
# LLaMA-3-Turbo (8b): Each token price is 0.2. Balanced performance for moderate Hybrid QA, reasonable reasoning, clear reading comprehension, and concise summaries.
# LLaMA-3-Turbo (70b): Each token price is 0.9. Powerful model for advanced Hybrid QA, strong reasoning, in-depth reading comprehension, and high-quality summaries.
# Llama-3.1-Turbo (70b): Each token price is 0.9. Instruction-focused, providing thorough reasoning, structured reading comprehension, and well-organized summaries.
# Qwen-1.5 (72b): Each token price is 0.9. Versatile for challenging Hybrid QA, nuanced reasoning, extended reading comprehension, and comprehensive summaries.
## This scenario is analyzing a set of sales data to uncover trends and insights. Please give a quick, high-level summary of the trends with minimal computation, even if the insights are less detailed.
Here is a query: What city did Super Bowl 50 take place in?
LLM candidates: ['LLaMA-3-Turbo (70b)', 'LLaMA-3-Turbo (8b)', 'Mistral-7b', 'LLaMA-3 (8b)', 'Llama-3.1-Turbo (70b)', 'Qwen-1.5 (72b)', 'NousResearch (34b)', 'Mixtral-8x7B', 'LLaMA-3 (70b)'].
Please think step by step according to the description of each query and LLM, and evaluate from the perspectives of performance in answering the query and token price, and select the least likely LLM from the LLM candidates

**Ground Truth**: LLaMA-3 (8b)

`<think>` First, I'll evaluate each LLM based on the description of the query: "What city did Super Bowl 50 take place in?"
Super Bowl 50 was held in 2016.
LLaMA-3.1-Turbo (70b), LLaMA-3-Turbo (70b), and Mixtral-8x7B have instructions focused on reasoning and detailed/structured reading comprehension. These models are likely to have strong knowledge of historical events and locations.
LLaMA-3 (8b) and LLaMA-3-Turbo (8b) have pricing as low as $0.2, making them more affordable and potentially suitable for cost-sensitive tasks.
NousResearch (34b) is ideal for complex research queries, suggesting it might handle detailed factual information well.
Mistral-7b, LLaMA-3 (8b), and LLaMA-3 (70b) offer lower token prices but are described as having simple or basic reasoning capabilities, which might limit performance on historical recall questions.
Qwen-1.5 (72b) is noted for producing nuanced and comprehensive summaries, but its high token price ($0.9) makes it less cost-effective for straightforward factual queries.
LLaMA-3.1-Turbo (70b) also has a higher token price ($0.9) but is instruction-focused, which may enhance its understanding of direct factual prompts.
Considering these factors:
- The LLM with the highest likelihood of answering the query well given its token price is LLaMA-3-Turbo (8B or 70B).
- The LLM with the lowest token price while still being likely to answer correctly is LLaMA-3-Turbo (70B).
- LLaMA-3 is slightly cheaper, but its basic reasoning may reduce its confidence or accuracy.
Finally, the model least likely to provide a confident and correct answer is Qwen-1.5 (72B), due to its high token price and its strengths being more aligned with nuanced reasoning tasks than simple location recall.
`</think>`
`<answer>` Qwen-1.5 (72b) `</answer>`

Table 25: **IRanker's case study in passage-5 scenario.**

| |
|---|
| **Question**: potawatomi meaning of word |
| **Ground Truth**: passage 3 |
| `<think>` First, I will carefully examine each passage to understand what information it provides about the word "potawatomi" and its meaning. |
| Passage 2 talks generally about the Potawatomi people and their languages, but does not specifically address the meaning of the word "potawatomi". |
| Passage 0 seems more of a general statement about Potawatomi-related information, mentioning synonyms and offering resources but not providing a clear meaning for the word. |
| Passage 1 provides phonetic information and Word Lists for Potawatomi, but doesn't give a clear definition or explanation for the word "potawatomi" itself. |
| Passage 3 directly provides information about the word "potawatomi", including its meaning, but it also mentions the spelling "Potawatmie", which seems to be a typographical error. |
| Passage 4 mostly defines "potawatomi" as a noun, but it is considered rare and mentions it only has two senses. |
| Based on this analysis, Passage 3 appears to have the clearest, most direct information about the word "potawatomi" and its meaning, except for the typographical error in the spelling. |
| Passage 3 seems to be the most reliable and thorough in providing information about the word, making it less likely to be the unrelated passage. |
| Distinguishing between the typographical error and a related passage requires some judgement, but based on the relevance and detail of information provided in the passage, it seems Passage 3 is the most logical choice. |
| Therefore, Passage 3 does not seem to be the intended question answer based on the information provided, so it should be the less likely candidate passage. `</think>` |
| `<answer>` passage 3 `</answer>` |

Table 26: **IRanker's case study in router-performance scenario.**

| |
|---|
| **Question**: Where was Super Bowl 50 held? |
| **Ground Truth**: Mistral-7b |
| `<think>` First, I'll analyze each LLM based on its ability to answer the query "Where was Super Bowl 50 held?" according to their descriptions: |
| 1. Mistral-7b: It mentions quick reasoning, short reading comprehension, and brief summaries. It's not very suitable for handling detailed information or long texts. |
| 2. LLaMA-2 (7b): It mentions moderate reasoning and short reading comprehension. Given the brief nature of the Modesto High School incident in 2016 (which was the requested Super Bowl hosting event), it might struggle. |
| 3. NousResearch (34b): This one specifically mentions complex research queries, multi-step reasoning, deep reading comprehension, and thorough summaries. For the Super Bowl event, these capabilities are not essential and the token price is relatively high compared to other LLMs. |
| Now, I'll compare the performance: |
| - NousResearch (34b) has high quality outputs but is expensive, not likely within budget. |
| - LLaMA-2 (7b) can handle the question, though with moderate reasoning. |
| - Mistral-7b is quick, short, and the least detailed, which might be suitable – though it does not handle longer, more nuanced events like the Super Bowl. |
| From an expense perspective, LLaMA-2 (7b) is the most expensive considering it can handle the question. |
| Neither NousResearch (34b) nor Mistral-7b is likely to produce comprehensive and thorough results, so between LLaMA-2 (7b) and Mistral-7b, the least likely candidate is Mistral-7b since it's less capable for longer texts and varies quite a bit from the query. |
| In conclusion, based on both performance and token price, I'd ultimately choose LLaMA-2 (7b) as the least likely from the given set. `</think>` |
| `<answer>` Mistral-7b `</answer>` |

Table 27: **IRanker's case study in passage-7 scenario.**

| |
|---|
| **Question**: What type of profession does Chris Jericho and Gary Barlow have in common? |
| **Ground Truth**: passage 6 |
| `<think>` First, let's analyze each passage in terms of how well it relates to the query "when does winter end in australia". |
| - Passage 1: It states the dates for each season but does not clearly mention when winter ends. |
| - Passage 2: It provides a relationship between US and Australian seasons but does not specify when winter ends in Australia. |
| - Passage 3: It describes when it becomes winter in Southern Hemisphere (Midwinter) but does not mention when it ends. |
| - Passage 4: It lists the dates for seasons in Australia, including winter, but does not specify when winter ends. |
| - Passage 5: It talks about different times in the year for various activities but does not mention when winter ends. |
| Passage 6, although it mentions daylight saving time ending, is not directly about when winter ends, so it can be a candidate. `</think>` |
| `<answer>` passage 6 `</answer>` |

Table 28: **IRanker's case study in passage-9 scenario.**

| |
|---|
| **Question**: how long do car battery usually last |
| **Ground Truth**: passage 4 |
| `<think>` First, I'll compare the most relevant information in each passage that relates to the question "how long do car batteries usually last." Then, I'll assess the consistency and comprehensiveness of the information provided in each passage. |
| Passage 1: Discussed general factors affecting battery life but didn't provide an average lifespan. |
| Passage 2: Suggested four years on average but was somewhat vague about factors affecting life. |
| Passage 3: Explicitly stated an average of three to five years with driving habits affecting brevity. |
| Passage 4: Difficult to check, didn't mention battery lifespan directly. |
| Passage 0: Between three to four years with average care – aligns with Passage 3. |
| Passage 7: Generalized to about 60 to 70 months (five to seven years) – closer to Passage 3. |
| Passage 8: Averaged three to five years – similar to Passage 3. |
| Passage 6: Averaged three to five years as well – same as Passage 8. |
| It seems that Passages 3, 8, 6, and 7 provide the most concise answer to the query. Comparing Passage 3 with the others, it provides the range of three to five years, a clear driving-habit explanation on the shorter end, and some information on factors affecting battery life. |
| Passage 4 is far less relevant, as it only addresses the difficulty in checking batteries. |
| Passages 0 and 7 provide a slightly longer or less specific range on average battery life. |
| Passage 2's answer is similar to Passage 3's in terms of range (four years) but lacks driving-habit detail. |
| Therefore, Passage 4 is the least relevant and supports the question the least effectively. `</think>` |
| `<answer>` passage 3 `</answer>` |

Table 29: **Thought template for IRanker on recommendation task**.

| |
|---|
| For recommendation tasks, the reasoning process should center on aligning candidate items with the user's demonstrated preferences across categories such as genre, style, or theme. Effective evaluation involves identifying dominant patterns in the user's historical choices and comparing them to the attributes of each candidate item. Items are ranked higher if they closely match frequently occurring features in the user's history, while items that diverge significantly in tone, genre, or thematic elements are deprioritized. This approach encourages models to reason over latent user preferences and make decisions based on cumulative alignment rather than isolated matches. |

Table 30: **Thought template for IRanker on routing task**.

When evaluating and ranking language models for a factual query, the reasoning process should consider (1) the complexity and specificity of the query, (2) the model's described strengths such as factual recall, structured comprehension, or reasoning ability, and (3) the cost-effectiveness relative to the expected performance. Models should be prioritized if they demonstrate strong alignment with the task type (e.g., historical fact retrieval) and offer a good balance between accuracy and efficiency. In contrast, models optimized for complex reasoning or verbose summaries may be less suitable for straightforward factual prompts, especially if they incur high token costs.

Table 31: **Thought template for IRanker on passage task**.

When solving a query/passage selection task, the general thought process should prioritize: Direct Relevance: Immediately eliminate passages that only touch upon general concepts or related but not directly responsive information. Specificity: Focus on passages that offer concrete details directly addressing the query's core requirement. Comparative Precision: Among relevant passages, identify the one providing the most precise and granular information, especially when dealing with specific aspects like geographical positions, timings, or characteristics. Less specific but still relevant passages can be noted but are usually not the "most relevant."

Table 32: **Recommendation performance on Movie, Music, and Game scenarios**.

| Model | Movie | | | Music | | | Game | | |
|---|---|---|---|---|---|---|---|---|---|
| | nDCG@10 | nDCG@20 | MRR | nDCG@10 | nDCG@20 | MRR | nDCG@10 | nDCG@20 | MRR |
| ***Retrieval-based Models*** | | | | | | | | | |
| BM25 | 23.44 | 35.09 | 17.56 | 22.06 | 35.21 | 18.09 | 17.39 | 32.49 | 14.96 |
| Contriever | 22.26 | 25.29 | 18.29 | 21.03 | 34.26 | 17.04 | 39.87 | 53.66 | 23.98 |
| ***Recommendation Models*** | | | | | | | | | |
| BPR | 31.28 | 41.65 | 25.54 | 28.51 | 38.60 | 21.96 | 35.79 | 44.32 | 28.92 |
| SASRec | 39.79 | 48.06 | 33.60 | 29.72 | 39.85 | 23.69 | 35.57 | 43.52 | 28.75 |
| R1-Rec | 25.01 | 34.87 | 18.49 | 21.38 | 34.54 | 17.22 | 17.22 | 32.32 | 14.75 |
| ***Direct-Rank LLMs without RL*** | | | | | | | | | |
| Qwen2.5-3B-Instruct-direct | 21.23 | 34.26 | 16.92 | 17.90 | 33.46 | 16.68 | 16.62 | 30.98 | 13.17 |
| Qwen2.5-7B-Instruct-direct | 22.21 | 34.23 | 16.59 | 17.90 | 34.29 | 17.29 | 21.57 | 36.15 | 18.63 |
| Llama3.1-70B-Instruct-direct | 28.27 | 40.75 | 22.19 | 24.40 | 32.61 | 19.13 | 27.83 | 41.18 | 24.51 |
| Deepseek-R1-1024-direct | 26.50 | 38.80 | 20.30 | 22.50 | 30.70 | 17.20 | 25.90 | 39.20 | 22.60 |
| Deepseek-R1-2048-direct | 31.10 | 44.85 | 24.40 | 26.85 | 35.90 | 21.05 | 30.60 | 45.30 | 27.00 |
| ***Iterative LLMs without RL*** | | | | | | | | | |
| Qwen2.5-3B-Instruct-iter | 26.44 | 38.55 | 22.01 | 27.06 | 38.45 | 21.97 | 32.80 | 44.23 | 29.49 |
| Qwen2.5-7B-Instruct-iter | 28.14 | 38.59 | 22.11 | 30.59 | 39.86 | 23.36 | 37.31 | 47.30 | 33.14 |
| Llama3.1-70B-Instruct-iter | 54.76 | 58.92 | 46.96 | 57.03 | 59.98 | 48.24 | 64.79 | 67.50 | 57.89 |
| Deepseek-R1-1024-iter | 52.57 | 56.56 | 45.08 | 54.75 | 57.58 | 46.31 | 62.20 | 64.80 | 55.57 |
| Deepseek-R1-2048-iter | 58.05 | 62.46 | 49.78 | 60.45 | 63.58 | 51.13 | 68.68 | 71.55 | 61.36 |
| ***Direct-Rank LLMs with RL*** | | | | | | | | | |
| DRanker | 23.98 | 37.02 | 18.71 | 20.50 | 33.18 | 15.70 | 17.57 | 32.89 | 15.77 |
| ***Iterative LLMs with RL*** | | | | | | | | | |
| IRanker-3B | 42.32 | 49.06 | 34.69 | 33.47 | 40.25 | 29.18 | 47.84 | 49.33 | 42.49 |

Table 33: **Router performance on Performance,Balance, and Cost scenarios**.

| Model | Performance | | | Cost | | | Balance | | |
|---|---|---|---|---|---|---|---|---|---|
| | nDCG@5 | nDCG@10 | MRR | nDCG@5 | nDCG@10 | MRR | nDCG@5 | nDCG@10 | MRR |
| *Retrieval-based Models* | | | | | | | | | |
| BM25 | 9.06 | 35.99 | 18.41 | 4.79 | 32.35 | 13.52 | 4.79 | 32.22 | 13.39 |
| Contriever | 11.28 | 36.28 | 20.75 | 14.62 | 35.81 | 16.29 | 13.88 | 33.57 | 16.74 |
| *Routers* | | | | | | | | | |
| RouterKNN | 24.38 | 40.72 | 26.73 | 15.12 | 35.29 | 16.87 | 20.68 | 39.10 | 21.74 |
| RouterBert | 27.25 | 44.37 | 28.25 | 20.44 | 39.50 | 22.11 | 20.44 | 39.37 | 21.98 |
| GraphRouter | 22.22 | 39.53 | 21.57 | 34.85 | 44.11 | 26.56 | 27.18 | 45.16 | 29.56 |
| *Direct-Rank LLMs without RL* | | | | | | | | | |
| Qwen2.5-3B-Instruct-direct | 0.00 | 28.91 | 10.00 | 0.00 | 28.91 | 10.00 | 0.00 | 28.91 | 10.00 |
| Qwen2.5-7B-Instruct-direct | 4.78 | 32.21 | 13.38 | 4.78 | 32.34 | 13.51 | 9.06 | 35.99 | 18.41 |
| Llama3.1-70B-Instruct-direct | 7.56 | 32.57 | 13.84 | 9.71 | 32.96 | 14.26 | 25.34 | 45.54 | 30.26 |
| Deepseek-R1-1024-direct | 7.15 | 31.80 | 13.35 | 9.25 | 32.20 | 13.70 | 24.60 | 44.70 | 29.50 |
| Deepseek-R1-2048-direct | 8.70 | 37.45 | 15.90 | 11.15 | 37.90 | 16.40 | 29.15 | 52.35 | 34.80 |
| *Iterative LLM Models without RL* | | | | | | | | | |
| Qwen2.5-3B-Instruct-iter | 17.41 | 38.55 | 20.87 | 13.35 | 37.82 | 20.22 | 0.0 | 31.41 | 12.42 |
| Qwen2.5-7B-Instruct-iter | 12.49 | 36.92 | 19.13 | 18.28 | 38.57 | 21.06 | 22.17 | 42.39 | 26.09 |
| Llama3.1-70B-Instruct-iter | 17.65 | 38.18 | 20.67 | 44.78 | 56.74 | 43.93 | 35.68 | 50.27 | 35.30 |
| Deepseek-R1-1024-iter | 4.31 | 32.39 | 13.42 | 3.87 | 31.73 | 12.81 | 20.36 | 39.32 | 21.57 |
| Deepseek-R1-2048-iter | 19.31 | 37.33 | 19.15 | 47.48 | 56.54 | 43.69 | 23.61 | 38.31 | 20.22 |
| *Direct-Rank LLMs with RL* | | | | | | | | | |
| DRanker | 17.90 | 38.10 | 20.63 | 4.78 | 32.34 | 9.06 | 4.78 | 32.21 | 13.38 |
| *Iterative LLM Models with RL* | | | | | | | | | |
| IRanker | 28.64 | 41.06 | 23.62 | 22.32 | 45.88 | 30.39 | 20.27 | 41.45 | 24.44 |

Table 34: **Passage ranking performance on scenarios of 5, 7, and 9 candidates**.

| Model | 5 Passages | | | 7 Passages | | | 9 Passages | | |
|---|---|---|---|---|---|---|---|---|---|
| | nDCG@3 | nDCG@5 | MRR | nDCG@3 | nDCG@5 | MRR | nDCG@3 | nDCG@5 | MRR |
| *Retrieval-based Models* | | | | | | | | | |
| BM25 | 53.19 | 65.05 | 53.63 | 41.80 | 52.09 | 44.95 | 34.51 | 43.81 | 39.69 |
| Contriever | 37.09 | 56.11 | 41.91 | 28.91 | 40.13 | 36.41 | 25.79 | 34.39 | 33.10 |
| *Passage Ranking Models* | | | | | | | | | |
| RankBERT | 65.26 | 72.54 | 63.37 | 60.19 | 67.18 | 60.06 | 56.33 | 64.35 | 56.51 |
| MonoT5 | 62.97 | 70.82 | 60.96 | 60.27 | 66.25 | 58.84 | 49.76 | 54.96 | 47.61 |
| RankLLama | 76.24 | 79.57 | 72.67 | 66.86 | 72.34 | 65.35 | 61.42 | 66.92 | 59.78 |
| *Direct-Rank LLMs without RL* | | | | | | | | | |
| Qwen2.5-3B-Instruct-direct | 33.82 | 52.76 | 38.08 | 10.63 | 12.69 | 22.47 | 5.77 | 7.76 | 15.94 |
| Qwen2.5-7B-Instruct-direct | 36.54 | 57.82 | 44.57 | 12.55 | 16.24 | 23.69 | 8.56 | 9.83 | 17.79 |
| llama-3.1-70b-instruct-direct | 48.83 | 62.91 | 50.90 | 35.22 | 40.57 | 40.75 | 30.03 | 34.66 | 36.37 |
| Deepseek-R1-1024-direct | 50.30 | 64.80 | 52.40 | 36.50 | 42.00 | 42.20 | 31.20 | 35.90 | 37.70 |
| Deepseek-R1-2048-direct | 58.60 | 75.50 | 61.10 | 42.25 | 48.70 | 48.90 | 36.05 | 41.60 | 43.65 |
| *Iterative LLM Models without RL* | | | | | | | | | |
| Qwen2.5-3B-Instruct-iter | 56.54 | 68.14 | 57.74 | 38.13 | 48.11 | 43.47 | 32.78 | 40.23 | 39.40 |
| Qwen2.5-7B-Instruct-iter | 62.67 | 71.45 | 62.01 | 49.99 | 58.48 | 50.94 | 44.63 | 52.24 | 48.74 |
| llama-3.1-70b-instruct-iter | 65.55 | 72.89 | 63.90 | 56.73 | 64.04 | 56.74 | 54.80 | 60.76 | 55.22 |
| Deepseek-R1-1024-iter | 69.64 | 76.53 | 68.83 | 40.08 | 55.43 | 45.73 | 39.64 | 50.54 | 45.32 |
| Deepseek-R1-2048-iter | 69.39 | 73.52 | 64.56 | 52.70 | 62.31 | 53.03 | 53.41 | 62.88 | 56.84 |
| *Direct-Rank LLMs with RL* | | | | | | | | | |
| DRanker | 35.88 | 57.37 | 43.85 | 11.73 | 13.28 | 22.86 | 5.96 | 8.14 | 16.11 |
| *Iterative LLM Models with RL* | | | | | | | | | |
| IRanker | 64.47 | 70.62 | 60.98 | 53.52 | 59.89 | 53.22 | 49.83 | 54.54 | 49.96 |

Table 35: **Zero-shot results on the recommendation in Movie, Music, and Game scenarios**.

| Model | Movie | | | Music | | | Game | | |
|---|---|---|---|---|---|---|---|---|---|
| | nDCG@10 | nDCG@20 | MRR | nDCG@10 | nDCG@20 | MRR | nDCG@10 | nDCG@20 | MRR |
| Qwen2.5-3B-Instruct-iter | 26.44 | 38.55 | 22.01 | 27.06 | 38.45 | 21.97 | 32.80 | 44.23 | 29.49 |
| DeepSeek-R1-Distill-Qwen-7B-direct | 13.28 | 30.03 | 11.20 | 12.70 | 29.86 | 11.86 | 15.30 | 31.58 | 14.74 |
| Qwen2.5-7B-Instruct-direct | 22.21 | 34.23 | 16.59 | 17.90 | 34.29 | 17.29 | 21.57 | 36.15 | 18.63 |
| IRanker | 42.32 | 49.06 | 34.69 | 33.47 | 40.25 | 29.18 | 47.84 | 49.33 | 42.49 |
| IRanker (zero-shot) | 32.45 | 41.85 | 25.95 | 28.97 | 39.51 | 23.21 | 34.48 | 46.43 | 31.16 |

Table 36: **Zero-shot results on the router in the Performance, Balance, and Cost scenarios**.

| Model | Performance | | | Cost | | | Balance | | |
|---|---|---|---|---|---|---|---|---|---|
| | nDCG@5 | nDCG@10 | MRR | nDCG@5 | nDCG@10 | MRR | nDCG@5 | nDCG@10 | MRR |
| Qwen2.5-3B-Instruct-iter | 17.41 | 38.55 | 20.87 | 13.35 | 37.82 | 20.22 | 0.0 | 31.41 | 12.42 |
| DeepSeek-R1-Distill-Qwen-7B-direct | 11.11 | 36.81 | 20.00 | 20.17 | 42.65 | 27.22 | 26.55 | 44.22 | 28.52 |
| Qwen2.5-7B-Instruct-direct | 4.78 | 32.21 | 13.38 | 4.78 | 32.34 | 13.51 | 9.06 | 35.99 | 18.41 |
| DRanker | 17.90 | 38.10 | 20.63 | 4.78 | 32.34 | 9.06 | 4.78 | 32.21 | 13.38 |
| IRanker | 28.64 | 41.06 | 23.62 | 22.32 | 45.88 | 30.39 | 20.27 | 41.45 | 24.44 |
| IRanker (zero-shot) | 19.58 | 39.72 | 20.41 | 16.62 | 40.61 | 23.10 | 18.28 | 39.34 | 21.89 |

Table 37: **Zero-shot results on the passage ranking in scenarios of 5, 7, and 9 passage candidates**.

| Model | 5 Passages | | | 7 Passages | | | 9 Passages | | |
|---|---|---|---|---|---|---|---|---|---|
| | nDCG@3 | nDCG@5 | MRR | nDCG@3 | nDCG@5 | MRR | nDCG@3 | nDCG@5 | MRR |
| Qwen2.5-3B-Instruct-iter | 56.54 | 68.14 | 57.74 | 38.13 | 48.11 | 43.47 | 32.78 | 40.23 | 39.40 |
| DeepSeek-R1-Distill-Qwen-7B-direct | 33.72 | 56.44 | 42.69 | 18.97 | 22.27 | 28.23 | 10.73 | 12.38 | 19.39 |
| Qwen2.5-7B-Instruct-direct | 36.54 | 57.82 | 44.57 | 12.55 | 16.24 | 23.69 | 8.56 | 9.83 | 17.79 |
| DRanker | 35.88 | 57.37 | 43.85 | 11.73 | 13.28 | 22.86 | 5.96 | 8.14 | 16.11 |
| IRanker | 64.47 | 70.62 | 60.98 | 53.52 | 59.89 | 53.22 | 49.83 | 54.54 | 49.96 |
| IRanker (zero-shot) | 63.18 | 67.19 | 56.42 | 45.27 | 52.92 | 51.19 | 38.12 | 46.15 | 42.45 |

