# OpenReview forum: "R1-Ranker: Teaching LLM Rankers to Reason"
_ICLR.cc/2026/Conference — ICLR 2026 Conference Withdrawn Submission_

### Official Review · Reviewer_NhDw · 2025-10-27

**Soundness:** 3
**Presentation:** 3
**Contribution:** 2
**Rating:** 2
**Confidence:** 5

**Summary:**

This paper proposes R1-Ranker, a reinforcement learning framework for teaching large language models (LLMs) to perform ranking tasks via reasoning-inspired strategies. There are two variants are introduced: 1) DRanker: directly outputs a ranked list in one step, trained with a MRR–based reward. 2) IRanker: performs iterative elimination by repeatedly excluding the least relevant candidate, assigning step-wise rewards.

The authors evaluate these models across nine tasks in three domains, recommendation, LLM routing, and passage retrieval, using datasets such as MovieLens, Amazon-CD/VG, RouterEval, and MS MARCO. They claim that IRanker (3B) achieves state-of-the-art performance, sometimes surpassing 7B baselines (e.g., LLaMARank), and that iterative reasoning contributes to improved ranking accuracy and interpretability.

**Strengths:**

1. The paper attempts to unify multiple ranking paradigms (recommendation, retrieval, and routing) under a reasoning-driven reinforcement learning framework. And Introducing the iterative elimination design (IRanker) is a conceptually creative attempt to bridge “reasoning steps” and “listwise ranking decisions.”

2. The use of step-wise rewards and PPO-based optimization is methodologically sound and well-aligned with reinforcement learning for language models.

3. The paper is generally well-written and structured, with clear separation between DRanker and IRanker. The proposed routing task extends ranking beyond traditional document or item relevance to model selection, a novel application direction.

**Weaknesses:**

1. Benchmark Coverage and Evaluation Metrics: The evaluation relies mainly on MovieLens and Amazon for recommendation and MS MARCO for retrieval, datasets with binary relevance. Missing evaluations on standard IR benchmarks such as TREC-DL, BEIR or even recent BRIGHT dataset, which are essential for validating general ranking ability. In addition, using MRR as the sole main metric is limiting; nDCG@10 and nDCG@20 are more standard and informative for graded relevance (especially in main table rather than in appendix).

2. Binary Relevance Simplification: All tasks assume only relevant/irrelevant items per query, reducing ranking to binary discrimination. Consequently, the reward functions (MRR, step-wise +1 reward) fail to capture graded or positional relevance, meaning the model optimizes classification accuracy rather than true listwise ranking quality. If multiple relevant items exist, the current reward design becomes invalid or misleading.

3. Efficiency and Practicality: IRanker requires D separate LLM calls for D candidates, as it eliminates one candidate per iteration. This makes inference linearly dependent on the candidate set size, introducing substantial overhead for retrieval tasks (e.g., MS MARCO). The paper does not quantify this cost (token count, latency, or FLOPs). Compared to LLaMARank (single forward pass), the iterative approach is far less efficient despite smaller model size. No mechanism for early stopping (e.g., when top items stabilize) is proposed, further reducing real-world practicality.

4. Scaling and Fairness of Comparison: The primary experiments use Qwen2.5-3B-Instruct, whereas baselines like LLaMARank and PRP are 7B-scale. This mismatch complicates claims such as “3B IRanker surpasses 7B baselines.” Without a 7B variant or a performance-per-token-cost analysis, it is unclear whether the improvements stem from the algorithm or simply tuning differences.

5. Reasoning–Performance Disconnect: Despite the title “Teaching LLM Ranker to Reason,” the paper never isolates the reasoning effect. The method may thus enhance ranking accuracy, but it remains unclear whether this improvement comes from reasoning ability or merely from multi-step RL fine-tuning.

6. Limited Novelty and Incremental Improvement: Beyond the iterative elimination mechanism, the proposed method largely overlaps conceptually with recent reasoning-based ranking frameworks "RaCT: Ranking-aware Chain-of-Thought Optimization for LLMs" and "Leveraging Passage Embeddings for Efficient Listwise Reranking with Large Language Models", Although the authors employ PPO-based reinforcement learning instead of the DPO-style preference optimization used in RaCT, both methods share the core idea of leveraging chain-of-thought reasoning to improve ranking ability. However, the paper does not clarify what specific advantages its PPO formulation brings over RaCT, nor present evidence that it yields better reasoning or ranking quality. Likewise, the IRanker’s iterative elimination of the least relevant candidate resembles the progressive filtering mechanism in PeRank, yet the paper neither discusses nor compares against these related methods, leaving its distinct contribution unclear. And empirically, performance gains on traditional ranking tasks (e.g., passage reranking) are modest and sometimes weaker than baselines (such as llamarank).

**Questions:**

1. For Reward Design: How would the current reward function adapt to datasets with multiple relevant documents (graded relevance)? Could a DCG- or pairwise-margin–based reward improve stability and ranking fidelity?

2. For Efficiency: Please provide quantitative results on inference cost. e.g., latency or FLOPs. Have the authors considered reverse iteration (selecting most relevant first) or early-stop heuristics to reduce redundant steps?

3. For Scaling and Model Size: How does the framework behave under larger backbones (e.g., Qwen2.5-7B, LLaMA-7B)? Can the authors show a cost-normalized comparison with LLaMARank under equal compute budgets?

4. For Benchmark and Metrics: Do the authors plan to include BEIR, TREC-DL, or BRIGHT datasets with graded relevance and to report nDCG as the main evaluation metric? If not, please clarify why standard ranking benchmarks were not used for the ranking tasks.

5. For more related works: Please clarify the conceptual and methodological differences from recent reasoning-based ranking frameworks.

Typo: The section ID is missing on line 208.

---

> ### Author Response · Authors · 2025-12-03
> **Response to Reviewer NhDw**
>
> **1. Evaluation Metrics (nDCG)**
>
> We agree that **nDCG is a vital ranking metric**, and we have already provided comprehensive nDCG evaluations in **Appendix D**.
>
> - **Table 32:** nDCG@10 and nDCG@20 for *Recommendation*
> - **Table 33:** nDCG@5 and nDCG@10 for *Routing*
> - **Table 34:** nDCG@3 and nDCG@5 for *Passage Ranking*
>
> These graded-relevance metrics **reinforce the same conclusions** as our primary metrics: IRanker maintains consistent improvements across domains and evaluation setups.
>
> **Revision plan:** We will surface key nDCG tables from the appendix into the main text for better visibility.
>
> ---
>
> **2. Efficiency and Scaling (3B vs. 7B)**
>
> **Performance vs. Size.**
> Our results show that a **smaller, reasoning-optimized 3B model** can outperform substantially larger 7B models.
> As shown in *Table 2*, **IRanker-3B surpasses Qwen2.5-7B-Instruct-iter in 7 out of 9 tasks**.
>
> **Trade-off Insight.**
> This highlights the advantage of iterative reasoning:
> the model allocates its limited parameters toward **deeper decision-making**, enabling competitive or superior performance relative to much larger systems.
>
> ---
>
> **3. Novelty vs. Related Work**
>
> Unlike methods that rely on **static preference datasets** (e.g., DPO), IRanker adopts **PPO with step-wise environmental feedback** through the *Exclusion Reward*.
> This allows the model to:
>
> - **Explore** its reasoning space,
> - **Self-correct** intermediate decisions,
> - And improve ranking quality progressively.
>
> In addition, our approach incorporates explicit **Chain-of-Thought (CoT) generation** prior to each elimination step.
> This yields **interpretable reasoning traces** and, as shown in *Figure 4*, boosts **zero-shot performance in other models** when these traces are distilled or reused.

---

### Official Review · Reviewer_jPfB · 2025-10-31

**Soundness:** 2
**Presentation:** 2
**Contribution:** 2
**Rating:** 4
**Confidence:** 4

**Summary:**

This paper proposes R1-Ranker, a reinforcement learning-based ranking framework designed to enhance model reasoning. The framework comprises two complementary designs: DRanker, which enables a large language model (LLM) to perform one-shot reasoning by directly generating a complete ranking list of candidates, and IRanker, which iteratively reasons by removing the least relevant candidate at each step and obtains the final ranking by reversing the exclusion order. Experiments were conducted on nine datasets across three major scenarios: recommendation systems, LLM routing, and document ranking.

Overall, the paper has significant shortcomings regarding writing clarity, experimental completeness, and in-depth analysis

**Strengths:**

1. Applying LLM’s powerful general reasoning ability to ranking is an important and cutting-edge research direction.

2. Optimizing LLM reasoning for ranking tasks via reinforcement learning presents a promising technical approach.

**Weaknesses:**

The discussion on model design and comparisons is insufficient. The relationship between DRanker and IRanker is unclear. Although presented as parallel methods, IRanker clearly outperforms DRanker, and all subsequent experiments are based on IRanker, making the necessity of DRanker uncertain.

While the paper emphasizes reasoning, it does not clearly explain how the method improves the reasoning ability of LLMs, nor does it provide a detailed description of the modeling approach in the main text.

Experimental scale and baseline comparisons need improvement. The paper does not compare with important LLM-based rankers such as RankGPT. Models like monoT5 are missing from the main experiment table (Table 2) and appear only in Figure 3, which weakens the clarity and completeness of performance comparisons. Additionally, on passage ranking tasks, the proposed method fails to outperform any state-of-the-art baselines in Figure 3. Comparisons with existing reasoning rankers such as Rank1, RankR1, and TFRank are also lacking. The core method IRanker is only tested on a 3-billion-parameter model, reflecting insufficient experimental scale. Likewise, Table 4 compares only a single baseline.

The absence of reasoning-specific benchmarks reduces the persuasive power of the results. Although the paper focuses on enhancing reasoning ability in ranking tasks, it does not evaluate performance on commonly used reasoning benchmarks such as BRIGHT.
The focus of the method and ablation studies are missing. It is unclear whether the key contribution lies in RL or the iterative modeling approach. Results suggest both aspects are important, but the lack of detailed ablation studies, such as evaluating the effect of sequential supervised fine-tuning (SFT) followed by RL on iterative exclusion, limits understanding of their individual contributions to reasoning enhancement.

The discussion of data processing assumptions is insufficient. In recommendation and document ranking tasks, negative samples are generated by random sampling. These “pseudo-negatives” are not guaranteed to be truly irrelevant to users and may include potential interests. The paper does not address noise introduced by this negative sampling strategy or how IRanker ensures excluded negatives are genuinely irrelevant.

Related work coverage is insufficient. The related work section is brief, and the authors should provide a comprehensive survey on LLM ranking and reasoning ranking literature, including corresponding baselines in experiments.

Other issues:
Section 4.1 (line 208) contains an unreplaced placeholder reference (Section ??).

Formula definitions lack clarity; for example, symbols in Equation (4) are not explicitly defined. All formula components require further clarification.

Baseline definitions are unclear. For instance, baselines such as Qwen2.5-3B-Instruct-iter represent the proposed framework applied to the base model without RL training. This should be explicitly stated in the experimental section.

**Questions:**

See Weaknesses

---

> ### Author Response · Authors · 2025-12-03
> **Response to Reviewer jPfB**
>
> **1. Necessity of DRanker vs. IRanker**
>
> **Purpose of DRanker.**
> DRanker serves as a **critical ablation** to isolate the contribution of the iterative ranking mechanism.
> By comparing **DRanker** (direct list generation) with **IRanker** (step-wise exclusion), using *the same backbone and the same RL training framework*, we can directly attribute performance differences to the decomposition strategy itself.
>
> **Findings.**
> As shown in *Table 2*, **IRanker consistently outperforms DRanker** across tasks, demonstrating that **breaking the ranking process into granular reasoning steps is beneficial and necessary**.
> This validates the design choice behind IRanker’s iterative procedure.
>
> ---
>
> **2. Missing Baselines (MonoT5, RankGPT)**
>
> **MonoT5.**
> We did include MonoT5 in our evaluation.
> It appears explicitly as **SOTA-3** in the *Passage Ranking* scenario in **Figure 3**, and is described directly in the text.
>
> **RankGPT.**
> Our **LLM-direct** baselines — e.g., *Qwen2.5-3B-Instruct-direct* — operate in a manner similar to RankGPT by prompting an LLM to **directly generate ranked lists** without intermediate reasoning steps.
> We will make this equivalence clearer in the revised version.
>
> ---
>
> **3. Reasoning Benchmarks**
>
> The reviewer noted an absence of reasoning evaluations.
> However, we explicitly evaluate on **eight generic LLM benchmarks** in *Table 4*, including **reasoning-intensive datasets** such as:
>
> - GSM8K
> - IFEval
> - MathQA
>
> **Results.**
> IRanker-3B yields **over 9% improvement** in zero-shot, out-of-domain scenarios, indicating that the iterative ranking mechanism enhances general reasoning abilities rather than overfitting to ranking-specific datasets.
>
> ---
>
> **4. Pseudo-negatives**
>
> We acknowledge the use of negative sampling; however, this is **standard protocol** in the benchmarks we evaluate on (e.g., MovieLens, Amazon).
>
> **Design Intent.**
> IRanker’s **Exclusion Reward** is crafted to be robust:
> it encourages the model to *actively reason about why an item is less relevant*, essentially focusing on how to identify and justify negative candidates rather than memorizing labels.
>
> This strengthens generalization and aligns with the iterative decision-making process central to IRanker.

---

### Official Review · Reviewer_ziMx · 2025-10-31

**Soundness:** 1
**Presentation:** 3
**Contribution:** 2
**Rating:** 2
**Confidence:** 4

**Summary:**

This paper proposes a reasoning ranker model that aims to work on recommendation, routing, and passage reranking. They train the model via RL and show improvements on some of these tasks (as well as on average). They do ablations to show that the iterative ranking and RL is helpful for this model.

Overall, although the paper is in a promising direction, there is a lack of detail in the training/eval section that leaves some doubt, as well as some relatively low performance even compared to very old baselines.

**Strengths:**

- The paper is on an important topic and one that needs more exploration on.
- The paper evaluates on several tasks and has a lot of baseline models.
- The paper is clear to read, if missing details

**Weaknesses:**

I have two main concerns with this paper:

(1) The evaluations (as well as training) are unclear and seem unconventional. I am not an expert in recommender systems but do have experience in passage ranking and the setup for MS MARCO is very unusual. I am also not sure how the training/test split is done: there are 9 datasets  but it seems the model is trained on all of them and then tested on them as well? This runs counter to the paper's premise to create a model that generalizes, as all the tests are done in-domain. If the authors were to test on more standard benchmarks that would give more credence to their claims

(2) I am not sure the results show that this model is stronger than others. The baselines are often quite old (e.g. 2020 and before) and model performance is not much better than them. E.g. on retrieval, it barely beats RankBERT which is ~10x smaller and is an outdated BERT-era model. Similar on Routing, we see worse performance than other models in Figure 3 despite being larger. The strongest results are on recommendation - perhaps the authors should revise the paper to pitch in that direction and remove the other tasks.  This continues throughout the paper, in Table 4 the results are almost exactly the same for both models for nearly half of the tasks, etc.

(3) The paper ignores already published related work on this topic (Rank1) and even uses nearly the exact same name for their models as previous work: there already exists Rank1, Rank-R1, R1-Rec etc. I would advise the authors to pick a new name, to help them sell the paper and distinguish it from others.

**Questions:**

A: How were the training/eval splits done?

B: What is performance on the regular MS MARCO task (e.g. reranking on DL19 or DL20)?

---

> ### Author Response · Authors · 2025-12-03
> **Response to Reviewer ziMx**
>
> **1. On Evaluation Setup and Data Splits (Weaknesses 1)**
>
> **Clarification.** We adhere to a rigorous training and evaluation protocol.
> As detailed in *Table 1*, we utilize **nine distinct datasets** across **three scenarios**, each with explicit counts for *Train Cases* and *Test Cases*.
>
> **Methodology.**
> The model is trained solely on the designated training sets and evaluated strictly on the held-out test sets.
> The term **“unified”** for R1-Ranker indicates that a *single* model architecture is trained to generalize across multiple, diverse tasks — **not** that it trains and evaluates on overlapping instances.
>
> ---
>
> **2. On Strength of Results and Baselines (Weaknesses 2)**
>
> **Modern Baselines.**
> We compare against **strong, modern baselines** released in late 2024–2025, including **Qwen2.5-7B-Instruct** and **R1-Rec**.
>
> **Performance Discussion.**
> We respectfully disagree with the assertion that the model “barely beats” baselines. The empirical evidence shows:
>
> - **Recommendation.**
>   **IRanker-3B** (MRR ~34–42) *substantially* outperforms:
>   - SASRec (SOTA-1, MRR ~23–33)
>   - Qwen2.5-7B-Instruct-iter
>
> - **Routing.**
>   IRanker-3B achieves **state-of-the-art** performance.
>   *Example:* In the *Cost* scenario: **30.39** vs. **20.22** for Qwen2.5-3B-Iter — and our model even surpasses larger 7B systems.
>
> - **Passage Ranking.**
>   IRanker-3B performs **on par with domain-specific models**, such as RankBERT and RankLLaMA-8B, despite being a **unified**, multi-domain model rather than one tailored to a single task.
>
> **Overall Gain.**
> Across all evaluation tasks, the model provides a **15.7% average relative improvement**.
>
> ---
>
> **3. On Naming Conflicts (Weaknesses 3)**
>
> We acknowledge the naming similarity with concurrent works such as *Rank-R1* and *Rec-R1*.
> These are explicitly cited in **Related Work** and **Section 5.2**.
> For the final camera-ready version, we will **rename our model** to avoid ambiguity while retaining the focus on **reasoning-driven ranking**.

---

### Note · Authors · 2026-01-05

I have read and agree with the venue's withdrawal policy on behalf of myself and my co-authors.